# Scoring Antarctic surface mass balance in climate models to refine future projections

Tessa Gorte[1], Jan T. M. Lenaerts[1], and Brooke Medley[2]

[1]Department of Atmospheric and Oceanic Sciences, University of Colorado Boulder
[2]Cryospheric Sciences Laboratory, National Aeronautics and Space Administration's Goddard Space Flight Center

**Abstract.** An increase of Antarctic Ice Sheet (AIS) surface mass balance (SMB) has the potential to mitigate future sea level rise that is driven by enhanced solid ice discharge from the ice sheet. For climate models, AIS SMB provides a difficult challenge, as it is highly susceptible to spatial, seasonal and interannual variability.

Here we use a reconstructed data set of AIS snow accumulation as "true" observational data, to evaluate the ability of the CMIP5 and CMIP6 suites of models in capturing the mean, trends, temporal variability and spatial variability in SMB over the historical period (1850-2000). This gives insight into which models are most reliable for predicting SMB into the future. We found that the best scoring models included the National Aeronautics and Space Administration (NASA) GISS model and the Max Planck Institute (MPI) für Meteorologie's model for CMIP5, and one of the Community Earth System Model v2 (CESM2) models and one MPI model for CMIP6.

Using a scoring system based on SMB mean value, trend, and temporal variability across the AIS, as well as spatial SMB variability, we selected a subset of the top 10th percentile of models to refine 21st century (2000-2100) AIS-integrated SMB projections to $2274 \pm 282$ Gt yr$^{-1}$, $2358 \pm 286$ Gt yr$^{-1}$, and $2495 \pm 291$ Gt yr$^{-1}$ for Representative Concentration Pathways (RCPs) 2.6, 4.5, and 8.5, respectively. We also reduced the spread in AIS-integrated mean SMB by 79%, 79%, and 74% in RCPs 2.6, 4.5, and 8.5, respectively.

Notably, we find that there is no improvement from CMIP5 to CMIP6 in overall score. In fact, CMIP6 performed slightly worse on average compared to CMIP5 at capturing the aforementioned SMB criteria. Our results also indicate that model performance scoring is affected by internal climate variability (particularly the spatial variability), which is illustrated by the fact that the range in overall score between ensemble members within the CESM1 Large Ensemble is comparable to the range in overall score between CESM1 model simulations within the CMIP5 model suite. We also find that a higher horizontal resolution does not yield to a conclusive improvement in score.

## 1 Introduction

Surface mass balance (SMB) is the rate of accumulation of mass on the surface of the ice sheet and is characterized predominantly by precipitation and sublimation, and also includes runoff and blowing snow terms (Lenaerts et al., 2019). We neglect blowing snow and runoff and estimate SMB as precipitation minus sublimation (Lenaerts et al., 2012). As SMB variability is

25 dominated by that of AIS precipitation, which is subject to high spatial and temporal variability (Bromwich et al., 2011), SMB is also highly variable from year to year (Monaghan and Bromwich, 2008).

Over longer ($\sim$100-1000 year) time scales, AIS SMB was assumed – until recently – to be relatively constant. Frezzotti et al. (2013) found that current SMB values are not anomalously high compared to the past $\sim$1000 years. Monaghan et al. (2006) found no discernible trend in AIS snowfall in the period 1957-2003. More recent studies, adding more annually-resolved SMB records covering the period 1800 to present and improving the spatial extrapolation, contested those earlier findings (Thomas
et al. (2017); Medley and Thomas (2019)). These studies found that, integrated over the AIS, SMB has been increasing at a rate of $0.4 \pm 0.1$ Gt yr$^{-2}$ over the last 200 years, although the trends show substantial regional variability. Several studies have provided additional evidence of regional variations in SMB trends, with strong SMB increase in some areas (Philippe et al. (2016); Thomas et al. (2015); Thomas et al. (2017)), and no SMB increase, or even SMB decrease, in other areas (Burgener
et al., 2013). Synoptic-scale variability induces a strong regional variability of SMB (Fyke et al. (2017); Marshall et al. (2017)). Additionally, as the atmosphere is projected to warm both globally and especially in the polar regions, the atmosphere is expected to be able to hold more moisture per the Clausius-Clapeyron relation. As such, SMB is expected to show an overall increase. In recent decades, this forced SMB response is undetectable due to the significant natural SMB variability (Previdi and Polvani, 2016). Teasing apart the forced response from natural SMB variability requires longer SMB time series – on the
order of centuries. In 2017, Thomas et al. (2017) found no significant SMB trend over the last 1000 years. In 2019, however, Medley and Thomas (2019) found that, over the past 200 years, there is a statistically significant SMB increase that can be derived from ice core measurements.

Despite its importance for AIS mass balance and global mean sea level, there are only few robust observations of SMB across the continent. A lack of regular spatial and temporal distribution of observations has led to many efforts to model SMB
using both regional and global climate models (RCMs and GCMs, respectively). Because the AIS is so large, predicting SMB out onto timescales from decades to centuries requires the use of GCMs (Gallée et al., 2013). Some GCMs have been shown to capture positive precipitation and SMB trends (Palerme et al. (2014); Lenaerts et al. (2016)), but many of those models tend to overestimate annual precipitation values likely due to poor representation of coastal topography as previous studies have shown this to be a significant factor in how precipitation is represented of the AIS (Genthon et al., 2009). This allows the atmospheric
moisture to penetrate too far inland and leads to excessive precipitation on much of the grounded AIS, while underestimating precipitation nearby the coasts (Palerme et al. (2017)). This inability to reproduce modern observations brings into question the models' ability to accurately project future changes.

While past research by Palerme et al. (2014) compared model output to observations using CloudSat and ERA-Interim, their observational data sets only spanned a short period (2006-2011). The limited climatology of AIS precipitation combined
with its highly temporally variable nature means that large limitations exist to enable a comparison. Barthel et al. (2019) investigated the Ice Sheet Model Intercomparison Project for CMIP6 to determine a recommendation of which models to use for ice sheet model forcings based on best captured current Antarctic climate relative to observations and their ability to project certain metrics into the future. The object of this paper is similar in that Barthel et al. (2019) use scoring criteria to refine model selection specifically for ice sheet model forcing. Their work differs in that their criteria look more at the large-scale

circulation patterns around ice sheets and the data set to which they compare models consists of large-scale fields reanalysis fields. Additionally, they don't then use this subselection of models to constrain future projections. In this work, we use a data set that specifically accounts for AIS SMB using recent advancements in synthesizing ice cores and reanalysis products. These reconstructed data sets now allow for a new avenue to investigate the ability of GCMs to capture SMB into the more distant past (Medley and Thomas, 2019) – an avenue that we leverage for climate model evaluation of AIS SMB to compare the suite of CMIP5 and CMIP6 climate models to this new SMB reconstruction.

## 2 Data

### 2.1 SMB Reconstructions

To improve upon model estimates, several groups have combined ice core data with models to create spatio-temporally robust SMB data sets (Monaghan et al. (2006), Thomas et al. (2017), Medley and Thomas (2019)). In this paper, we use the AIS SMB reconstruction generated by Medley and Thomas (2019). The authors synthesize SMB time series from an extensive ice-core database with reanalysis-derived spatial coherence patterns to generate a continent-wide AIS SMB data set. While Medley and Thomas (2019) compared three reanalysis products, they also show that MERRA-2 performed better than the other two reconstructed products in matching observations. As such, we will use the MERRA-2 based data set provided by Medley and Thomas (2019) as a proxy for all three reconstructions and refer to it as "reconstruction."

The reconstructed uncertainty used throughout this paper is a combination of the observational uncertainty (i.e. uncertainty from the ice-core records) and inta-ensemble variability (eq. 1). The inclusion of the intra-ensemble variability takes into account the internal variability of climate models. See Supplementary Material for an explanation on this process.

$$recon.uncertainty = \sqrt{(obs.uncertainty)^2 + (intra-ensemblevariability)^2} \qquad (1)$$

For this work, we investigate AIS SMB in GCMs. GCMs have, compared to RCMs, relatively low horizontal resolution, which makes it difficult for them to reproduce the detailed AIS SMB. RCMs have been shown to be more accurate in capturing AIS SMB (Agosta et al., 2019); however, due to their high resolution, RCMs are also relatively computationally expensive to run for long periods ($\sim$100s of years). Because one of the goals of this paper is to investigate the future of SMB over Antarctica, we analyze GCMs for their ability to simulate these long-term climate effects. As RCMs are by definition regional, they need boundary forcings, which adds an additional layer of complexity and a source of uncertainty to running RCMs into the long-term future. An additional reason we choose to analyze GCMs is simply to figure out which GCMs perform best at capturing these SMB phenomena. There has been extensive work investigating SMB in RCMs (e.g., Agosta et al. (2019); van Wessem et al. (2017); Lenaerts et al. (2012)), but comparably little looking at GCMs. To investigate the global coupled response to future SMB changes, one needs GCMs. As such, this work is aimed to inform the modeling community who is interested in global ramifications of changing AIS mass balance, and the ice sheet modeling community who needs AIS SMB input for running dynamical ice sheet models (Seroussi et al., 2019). Several recent studies, such as Barthel et al. (2019), Krinner et al.

(2014), and Beaumet et al. (2019) have investigated the impacts of thermodynamical phenomena such as sea level pressure, zonal wind speed, and near-surface temperatures as well as phenomena like sea ice extent on AIS SMB, but have not scored climate models on their performance on SMB specifically. Here, we develop scoring criteria that assess AIS SMB exclusively, and focus less on the mechanisms behind SMB variability and change. To get a comprehensive look at how well global climate models capture SMB, we compared the suites of CMIP5 and CMIP6 models to the reconstruction.

## 2.2 Climate Models

We used all applicable CMIP5 and CMIP6 model outputs, of which there were 81 models and 42 independent models (i.e. different model physics and/or resolutions) respectively, for the historical simulations (1850-2005). For the future simulations, we only had available output for 30 CMIP5 models, 19 of which are independent, and 24 CMIP6 models, of which 16 are independent. See Tables 1-3 in Supplementary Material for a list of models and their resolutions. The future simulations include three different forcing scenarios for CMIP5: Representative Concentration Pathway (RCP) 2.6, RCP4.5, and RCP8.5. RCP2.6 represents a low emission scenario, RCP4.5 a mid-range emission scenario, and RCP8.5 a high emission scenario through the 21st century (van Vuuren et al., 2011) as well as three comparable forcing scenarios for CMIP6: Shared Socioeconomic Pathway (SSP) 1-2.6, SSP2-4.5, and SSP5-8.5 (Riahi et al., 2017).

We downloaded CMIP5 and CMIP6 precipitation and evaporation/sublimation output at monthly time resolution and, after calculating SMB as precipitation - evaporation/sublimation, converted an annual time scale and integrated across the grounded AIS using the Ice Sheet Mass Balance Inter-comparison Exercise Team's (IMBIE Team) ice sheet mask (Shepherd et al., 2018).

## 3 Methods

We formulated five criteria on which to score the historical runs of the models. Three of the criteria are based on the AIS-integrated SMB: mean, trends, variability – and two are based on AIS SMB spatial patterns: modes of SMB variability, and variance explained by these modes. As the models' abilities to capture SMB are presented in the format of a "score card," judging the models against each criterion will be hereinafter referred to as "scoring". These criteria were determined having in mind the following questions: (1) do the models adequately simulate several SMB observed characteristics in the recent past, and (2) are the models that perform well adequately simulating SMB for the right reasons? All five criteria are weighted equally in the final scoring to prevent the final score from being skewed by any given criterion.

### 3.1 AIS-integrated SMB criteria

To score the models based on AIS-integrated SMB, we took the mean SMB across the AIS for every year that the reconstruction overlapped the models (1850-2000) to generate a single 151-year, AIS-integrated time series. We then split the time series into three aspects: the mean value of the SMB time series values (mean value referring to the value obtained by integrating SMB over the entire AIS), the time series linear trend, and the time series interannual variability.

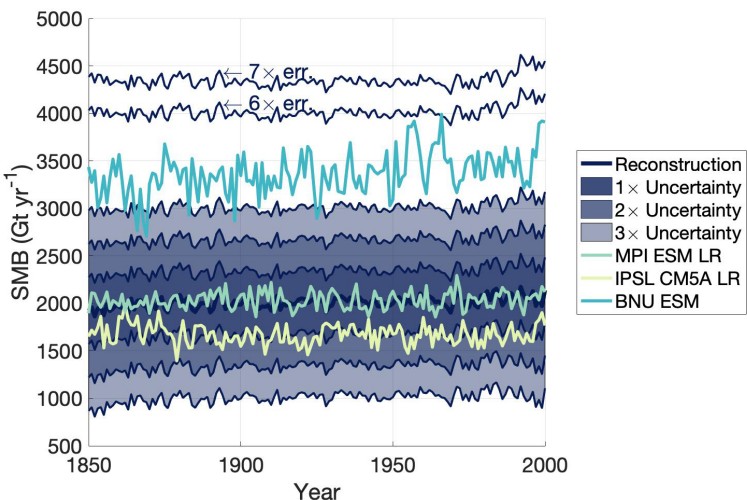

**Figure 1.** Time series of the reconstructed AIS-integrated SMB time series (dark indigo) with $1\times$, $2\times$, and $3\times$ the uncertainty in dark indigo, medium indigo, and light indigo, respectively. Three model AIS-integrated SMB time series, MPI ESM LR (green), IPSL CM5A LR (yellow), and BNU ESM (cyan) have been plotted as well to demonstrate different model scoring. MPI ESM LR is entirely captured within $1\times$ the reconstruction uncertainty and, thus, receives a score of 1. IPSL CM5A LR is entire captured within $2\times$ the uncertainty so its score for this criterion is 2. BNU ESM is fully captured within $7\times$ the uncertainty.

To score the time series mean value, we assigned a score, x, for how many x-times the reconstruction uncertainty was required for the entire time series to be within the reconstruction uncertainty. The minimum possible score, then, is one, for a model that represents SMB within $1\times$ the reconstruction uncertainty. Fig. 1 illustrates that a model that fits entirely within $1\times$ the reconstruction uncertainty (dark indigo) – MPI ESM LR – would receive a score of 1. A model that fits within $2\times$ the reconstruction uncertainty (medium indigo) – IPSL CM5A LR – would receive a score of 2. A poorer scoring model, BNU ESM, would receive a score of 6.

Similarly, for the time series trend, we assigned a score of x based on how many x-times the reconstructed trend uncertainty was required to capture the model trend. We looked at multiple time "slices" to investigate how well the models performed at capturing century-scale (100+ year) versus multi-decadal (50 year) SMB trends. To achieve this goal, we analyzed trends from 1850-2000, 1900-2000, and 1950-2000. The first two of these three time slices confirm the robustness of the trends with longer periods for trend analysis. The last time slice, 1950-2000, allows us to view SMB in the context of significant anthropogenic warming. However, the large interannual variability overwhelms the signal at shorter period lengths, which results in large uncertainty bounds. By looking at several time slices, we ensure consistency between the model and reconstruction over different intervals. It is equally important to confirm that pre-1950, the trends are relatively small. We performed a Monte Carlo simulation wherein we assumed a normal distribution where the standard deviation of the distribution is equal to the reconstruction uncertainty of possible SMB values for each year. We then created 10,000 potential SMB time series by choosing

SMB values based on that normal distribution for each year and recalculated the trend for each of these time series. Our uncertainty, then, was the standard deviation of this range of trends, similar to Medley and Thomas (2019).

For temporal variability, if a model should greatly underestimate the mean value, for example, the variability about that mean value will also likely be underestimated. To ensure that we are not double-counting the impact of SMB mean value (because this is already covered by the first scoring criterium), we calculated the variability about the normalized time series. To detrend and normalize each time series, then, to separate the SMB variability from its mean value, we performed the following analysis:

$$\text{normalized SMB} = \frac{\text{SMB} - \text{mean SMB}}{\text{mean SMB}}. \tag{2}$$

We then calculated the standard deviation of each time series and assigned a score, x, based on how many x-times the reanalysis standard deviation were required to capture the model standard deviation. For this criterion, we used the original MERRA-2 reanalysis precipitation minus evaporation data (1980-2019). Likely due to sampling only 53 ice core sites, the reconstruction produced a relatively low variability record. The reconstructed variability at any location can only be as large as the maximum variability in the ice cores. Thus, undersampling regions of stronger interannual variability will dampen the variability signal in the reconstruction. Analyses of the AIS-integrated SMB mean value and trend show that the reconstruction is generally in line with the literature (Medley and Thomas, 2019).

## 3.2  Spatial SMB criteria

To ensure model performance was not solely based on AIS-integrated SMB values, we also analyzed the spatial SMB variability. To do so, we performed an empirical orthogonal function (EOF) analysis on annual AIS SMB data from 1850-2000. EOF analysis maps the spatial pattern of a variable where the first mode represents the largest explained variance, the second mode - which is orthogonal to the first - represents the next largest explained variance, the third mode - which is orthogonal to both modes one and two - represents the third largest explained variance, and so on until all the variance is explained. By breaking this criterion down into two main factors, (1) spatial variability and (2) variance explained, both of which are considered as separate scoring criteria, we aim to determine the models' abilities to accurately capture the modes of variability as well as how much variance each EOF mode explained.

In the reconstruction, the top three modes of variability collectively explain roughly 76% of the total variance explained. The fourth mode explains only about 6% of the total variance and all other modes explain <5% of the total variance. As such, we only include the top three modes in our analysis. To avoid manually sorting the top three modes of variability for all 53 models, we generated difference maps between each of the top three reconstructed modes and each of the top three modes for each model: 9 difference maps for each model. For each grid point, we took the absolute value of the difference between the model and the reconstruction. We then summed those differences to generate a single number ("difference number") that represented the difference between the model and the reconstruction in terms of spatial variability. Mathematically, this looks like:

$$\text{difference number} = \sum_{lat} \sum_{lon} \left| \text{reconstruction}_{lat,lon} - \text{model}_{lat,lon} \right| \tag{3}$$

We did this for all nine combinations of model and reconstruction maps for the top three modes variability ($model_1$:reconstruction$_1$, $model_1$:reconstruction$_2$, $model_1$:reconstruction$_3$, $model_2$:reconstruction$_1$, $model_2$:reconstruction$_2$, etc.). For reconstruction mode 1 (reconstruction$_1$), then, we matched which model mode best represents this spatial variability by sorting the model modes based on the smallest difference number. We did this for each reconstruction mode (excluding previously matched model modes) to sort the modes based on the smallest difference. Summing the absolute value of these differences yielded a single number that explained how different a given model was from the reconstruction for each mode of variability. The score, then, for the variability of SMB is the total difference of all the top 3 modes.

Because the variance explained is also important for gauging how well models are performing at recreating the observed spatial patterns, we also summed the difference in variance explained for the top three sorted modes of variability for each model. Because the modes were sorted based on difference for the maps, each mode kept its variance explained to preserve the accuracy of the models regarding the dominance of each spatial pattern.

### 3.3 Final Scoring

After compiling scores for all five of the aforementioned scoring criteria, we removed any outliers by calculating the 1.5 quartile range of the data and neglecting models that fell outside of that range. We then normalized each set of scores to be on a scale from one to ten to ensure that each criterion was equally weighted. After this normalization, the outliers for any given criterion were retroactively assigned a score of ten for that criterion. The total score, then, is the average of all five sets of normalized scores. Because the scores are based on the difference between the reconstruction and the models, higher scores indicate poorer model performance.

To look at the impact of resolution and internal variability on the final scoring we correlated the horizontal resolution to final score and applied the same scoring analysis to the CESM Large Ensemble (CESM-LENS) experiment.

### 3.4 Future Projections

To reduce the uncertainty of for AIS SMB in the future, we created a subset of models that had a final score in the top 10th percentile (90th percentile and above) of CMIP5 and CMIP6. For our future projections, we investigated the impacts of SMB under forcing scenarios RCPs 2.6, 4.5, and 8.5 for CMIP5 and SSPs 1-2.6, 2-4.5, and 5-8.5 for CMIP6. We compared the top scoring models that could be projected out under the selected forcings (of which there are five: four for CMIP5 and one for CMIP6) to the entire scope of CMIP5 and CMIP6. We ran a Monte Carlo simulation in which five random models were selected 100,000 times. Those 100,000 sets of five random scores were compared to the five best scoring model scores using a two-sided t-test. From this, we found that, to a 95% confidence level, we can say that the five best scoring models are statistically significantly different from any random five CMIP5 or CMIP6 models.

Using this subset of best scoring models, we calculated the projected AIS-integrated mean value and trend in warming scenarios, RCPs 2.6, 4.5, and 8.5 and SSPs 1-2.6, 2-4.5, and 5-8.5, out to 2100. To see if and how the models respond differently

to different warming scenarios, we also calculated the AIS-integrated SMB sensitivity to temperature change as

$$200 \quad \text{Sensitivity} = \frac{\Delta\text{SMB}}{\Delta\text{T}}. \tag{4}$$

## 4 Results

The final overall scores are an unweighted average of all five different scores. After performing the analysis outlined in the Methods section, the top 90th percentile overall scoring models were determined to be GISS E2 H CC, GISS E2 R CC, GISS E2 R, MPI ESM LR, MPI ESM MR, and MPI ESM P from CMIP5, and CESM FV2 and MPI ESM2 LR from CMIP6. For comparison, these eight models have been added to Figures 3, 4, and 5, to show their performance in each scoring criterion relative to the rest of the CMIP model suites.

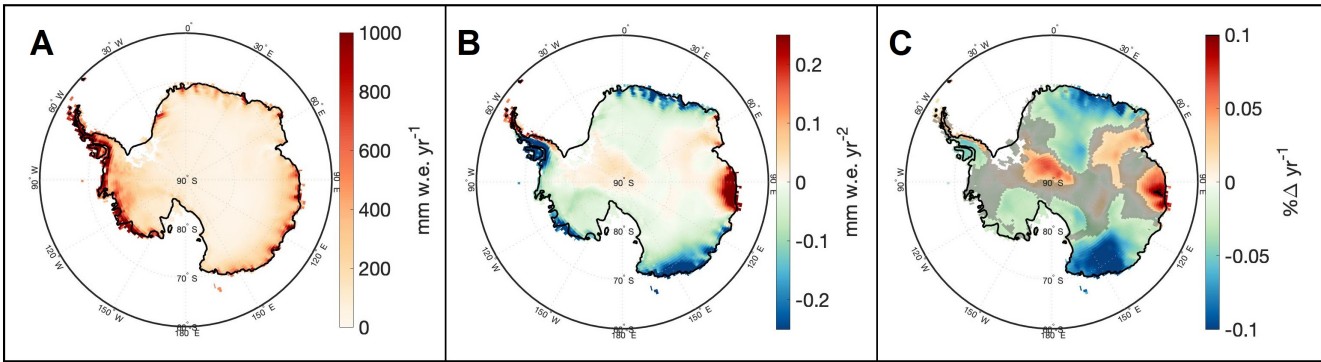

**Figure 2.** A spatial map of **A** the temporal average from 1801-2000 of the reconstructed AIS SMB, **B** the linear trend from 1801-2000 of the reconstructed AIS SMB, and **C** the relative SMB trend in percent SMB change per year. Non-shaded regions in panel **C** denote areas that are statistically significant.

Along with higher SMB values, the coastal regions of East Antarctica and the Antarctic Peninsula also show the highest absolute SMB trends in the reconstruction (Fig. 2**B**). The reconstruction also highlights large portions of East Antarctica as well as the Antarctic Peninsula as the regions with the most significant SMB trends from 1801-2000 (Fig. 2**C**). Taking the spatial average but keeping the temporal information yields the AIS-integrated, reconstructed SMB time series shown in Fig. 3**C** (black).

Panel (A) in Fig. 3 shows an example box plot for a suite of models in yellow and the reconstructed observations in black and grey. Panel (B) in Fig. 3 shows a box plot of the temporal average of the spatially integrated AIS SMB for CMIP5 and CMIP6. The interquartile range of AIS-integrated SMB in the CMIP5 models is between 1727 and 2282 Gt yr$^{-1}$, while the interquartile range in the CMIP6 models is between 1728 and 2196 Gt yr$^{-1}$. The best eight models range from 1909 to 2461 Gt yr$^{-1}$ for the temporal average AIS-integrated SMB mean value.

The reconstructed AIS SMB ranges from 1800 $\pm$ 338 Gt yr$^{-1}$ from 1850-1900 to 2039 $\pm$ 333 Gt yr$^{-1}$ from 1950-2000. All but one of the eight of best scoring models are fully captured within the reconstructed uncertainty for the entire 150 year

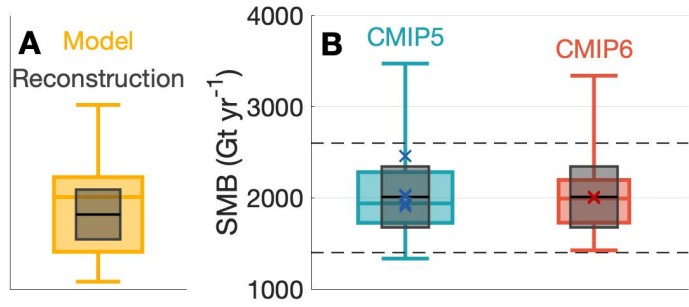

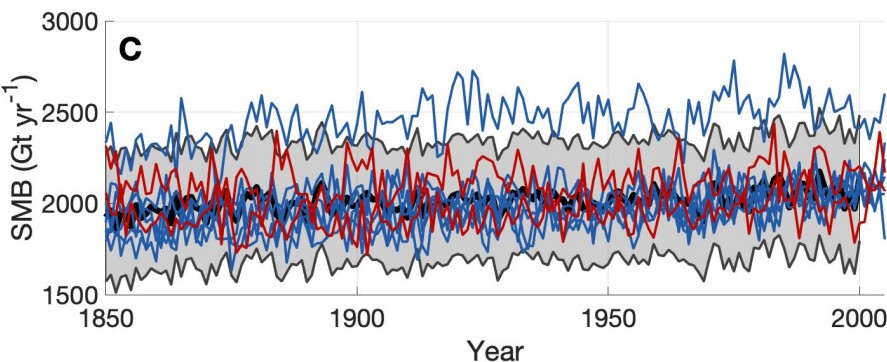

**Figure 3. A** An example of a box plot for model data (yellow) and reconstructed data (black and grey). The yellow shaded box shows the models' interquartile range while the whiskers extend to capture the entire distribution of modeled data. The line going through the box plot shows the median model value. The grey shaded box shows the reconstructed uncertainty around the reconstructed value shown as a black line. **B** A box plot of spatially integrated, temporally averaged (1850-2000) AIS SMB for CMIP5 (aqua) and CMIP6 (red). The dark blue x's associated with the CMIP5 box and the red x's associated with the CMIP6 box represent the eight best scoring models: GISS E2 H CC, GISS E2 R CC, GISS E2 R, MPI ESM LR, MPI ESM MR, and MPI ESM P from CMIP5 and CESM2 FV2 and MPI ESM2 LR from CMIP6. The black dashed lines indicate the lower and upper bounds of the time series plot in the bottom of Figure 3. **C** A time series of spatially integrated SMB for the reconstruction (black) and its uncertainty (shaded grey) with the best eight scoring models: GISS E2 H CC, GISS E2 R CC, GISS E2 R, MPI ESM LR, MPI ESM MR, and MPI ESM P from CMIP5 (dark blue) and CESM2 FV2 and MPI ESM2 LR from CMIP6 (red)

time series. The reconstruction and best scoring models all show generally increasing SMB from 1850-2000, albeit with large
interannual variability. Both the trend and variability are analyzed in follow-up evaluations and scoring.

While the reconstructed SMB time series and eight best scoring models show a generally increasing trend, the same is not true for all CMIP5 or CMIP6 models (Fig. 4). Looking at multiple time "slices" allows us to investigate if models capture the reconstructed SMB trends for the whole time series compared to more recent decades. Here, we looked at three time slices: the entire overlapping time series from 1850-2000, the last century from 1900-2000, and the last 50 years from 1950-2000.

The reconstructed linear SMB trends for the three time slices are $0.52 \pm 0.27$ Gt yr$^{-2}$ (1850-2000), $0.56 \pm 0.38$ Gt yr$^{-2}$ (1900-2000), and $1.0 \pm 1.3$ Gt yr$^{-2}$ (1950-2000). That implies that for all but the last time slice, 1950-2000, the reconstruction uncertainty trends are exclusively positive.

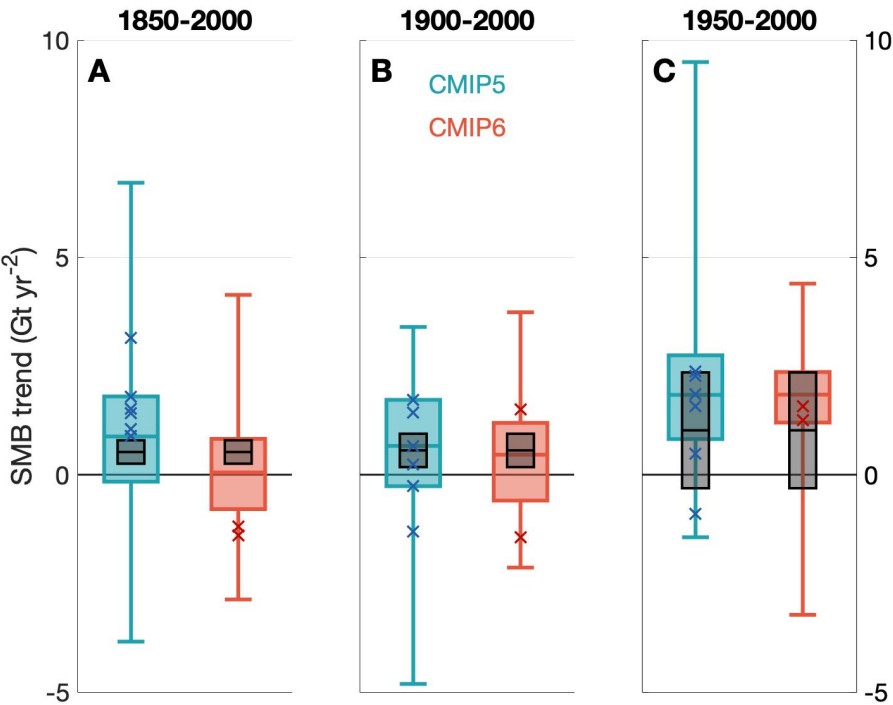

**Figure 4.** Box plots of the linear trends in spatially integrated AIS SMB in CMIP5 (blue) and CMIP6 (red) for the periods **A** from 1850 to 2000; **B** from 1900 to 2000 ; and **C** from 1950 to 2000. In all three panels, the grey boxes denote the reconstructed uncertainty around the reconstructed trend (black line). The eight best scoring models are represented by dark blue x's if they are among the CMIP5 suite of models or red x's if they are among the CMIP6 suite.

Looking at all of the CMIP5 and CMIP6 models, the median linear trend is positive for all three time slices and the trend interquartile ranges are from -0.8 to +1.8 Gt yr$^{-2}$ for 1850-2000, -0.6 to +1.7 Gt yr$^{-2}$ for 1900-2000, and 0.8 to +2.7 Gt yr$^{-2}$
for 1950-2000. For CMIP5, median trends for these time slices are 0.88 Gt yr$^{-2}$, 0.66 Gt yr$^{-2}$, and 1.8 Gt yr$^{-2}$ for 1850-2000, 1900-2000, and 1950-2000 respectively. For CMIP6, median trends for these time slices are 0.05 Gt yr$^{-2}$, 0.46 Gt yr$^{-2}$, and 1.8 Gt yr$^{-2}$ for 1850-2000, 1900-2000, and 1950-2000 respectively. The eight best scoring models range from -1.4 to +3.1 Gt yr$^{-2}$, -1.4 to +1.7 Gt yr$^{-2}$, and -0.9 to +2.4 Gt yr$^{-2}$ for the same respective time spans. The spread in the eight best scoring models reduces the total spread in AIS-integrated trend by 57%, 62%, and 70%, respectively. In both CMIP5 and CMIP6,

for the first two time slices, the reconstructed trend and uncertainty are captured within the interquartile range for all CMIP5 models. For 1950-2000, the models tend to overestimate the reconstructed trend.

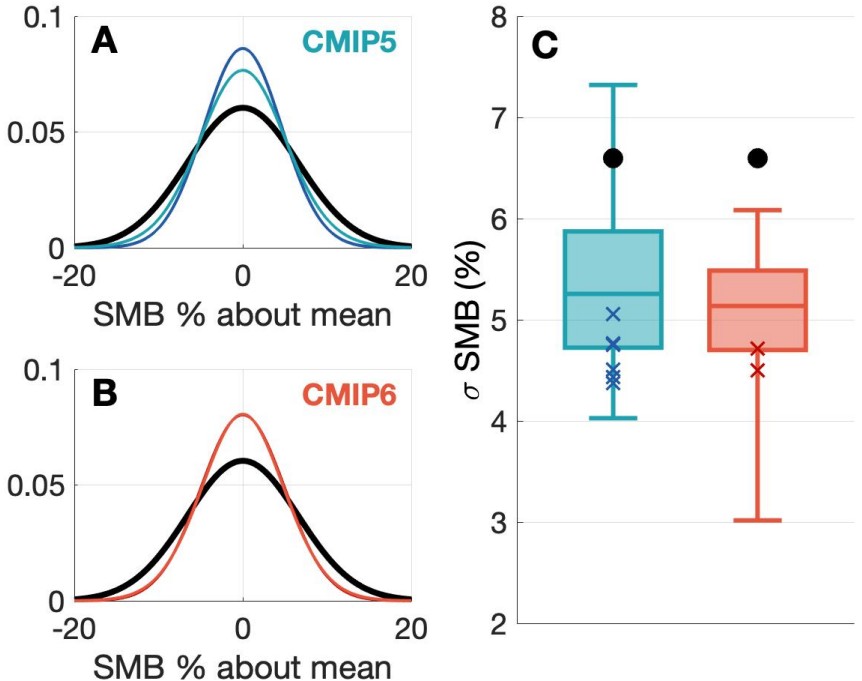

**Figure 5.** Gaussian distributions of SMB where the standard deviation is that of the SMB time series for the reconstruction (black) and **A** all CMIP5 models in light blue and the best scoring CMIP5 models in dark blue and **B** all CMIP6 models in light red and all CMIP6 models in red (the two Gaussians, here, are largely indistinguishable by eye as they overlap almost entirely). **C** Box plots of the CMIP5 (blue) and CMIP6 (red) SMB time series standard deviations. The black dots show the standard deviation of the original MERRA-2 reanalysis.

Apart from its trend magnitude and sign, SMB variability is also important for accurately representing SMB, and can be indicative of the relevant SMB driving mechanisms. Figure 5**A**, **B** shows the average detrended and normalized variability for CMIP5 and CMIP6 models as well as the reconstruction plotted as a normal distribution. The detrended and normalized
interannual variability in SMB in the reconstruction ranges between ∼-20% to 20%, while SMB in all the models varies between ∼-15 to 15%. Figure 5**C** shows a box plot the standard deviations of the normalized and detrended time series. The normalization process made it such that the standard deviations are calculated in % of variability about the mean value of the time series. The standard deviation for the normalized and detrended SMB in the reanalysis is about 6.6% compared to the best eight models which range between 4.4% to 5.1%. Most CMIP5 and CMIP6 models underestimate SMB variability. The
CMIP5 and CMIP6 models' standard deviations range from 4.0% to 7.3% and from 3.0% to 6.1%, respectively (Fig. 5**C**). For a summary of the ranges of the values for the three temporal criteria, see Table 1.

Just as temporal SMB variability is important for accurately capturing AIS SMB, spatial variations in SMB are also important in AIS SMB representation in models, as precipitation is not distributed uniformly. To look at the spatial variability in SMB, we performed EOF analysis and plotted looked at the top three modes of variability which collectively account for 76.3% of the total spatial variability.

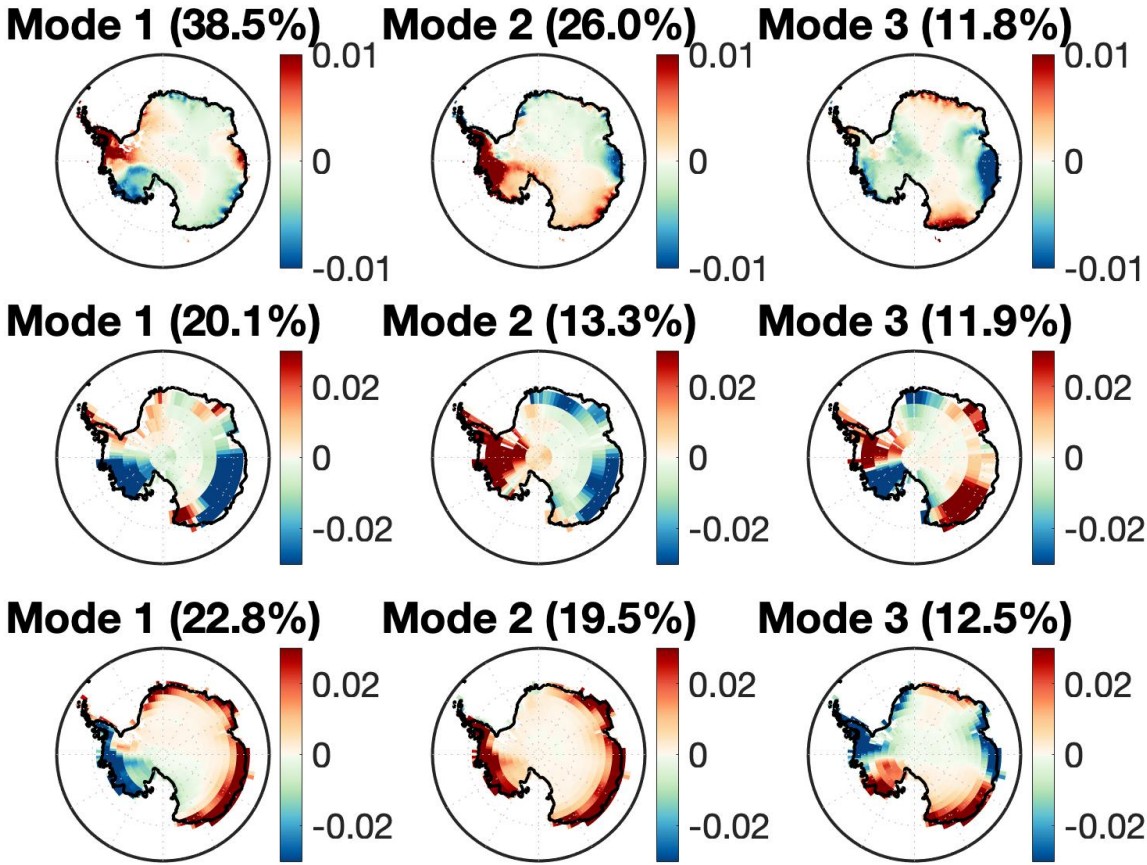

**Figure 6.** EOF analysis plots of the top 3 modes of variability for **A** the reconstruction, **B** a relatively high scoring model (CMCC CM), and **C** a low scoring model (CESM1 WACCM). Note that the scale for the model EOFs is 3× that of the reconstructed EOF.

Separated out, the top three modes of variability in the reconstruction from EOF analysis explain 39%, 26%, and 12% of the total variability, respectively (Fig. 6). High values on the EOF map indicate regions that explain large amounts of the variability in AIS SMB. The top mode of variability in the reconstruction shows a dipole pattern from the Antarctic Peninsula to the Ross Sea region. Mode 2 of the reconstruction EOF shows a strong signal over the entire Antarctic Peninsula and toward the Ross Ice Shelf region of West Antarctica. The third mode of variability shows a strong signal in Wilkes Land (East Antarctic region),

| Criterion (Time Span) | Reconstruction or Reanalysis | Top 90th Percentile Models | All CMIP5 Models | All CMIP6 Models |
|---|---|---|---|---|
| Mean Value (1850-2000) | 2010 +/- 334 Gt yr$^{-1}$ | 1909 - 2461 Gt yr$^{-1}$ | 1335 - 3472 Gt yr$^{-1}$ | 1976 - 2196 Gt yr$^{-1}$ |
| Trend (1850-2000) | 0.52 +/- 0.27 Gt yr$^{-2}$ | -1.40 - 3.40 Gt yr$^{-2}$ | -3.84 - 6.72 Gt yr$^{-2}$ | -2.87 - 4.14 Gt yr$^{-2}$ |
| Trend (1900-2000) | 0.56 +/- 0.38 Gt yr$^{-2}$ | -1.43 - 1.73 Gt yr$^{-2}$ | -4.81 - 3.40 Gt yr | -2.14 - 3.74 Gt yr$^{-2}$ |
| Trend (1950-2000) | 1.0 +/- 1.3 Gt yr$^{-2}$ | -0.89 - 2.38 Gt yr$^{-2}$ | -1.44 - 9.50 Gt yr$^{-2}$ | -3.22 - 4.40 Gt yr$^{-2}$ |
| Temporal Variability (1850-2000) | 6.6 +/- ??? Gt yr$^{-1}$ | 4.37 - 5.06 Gt yr$^{-1}$ | 4.03 - 7.32 Gt yr$^{-1}$ | 3.02 - 6.08 Gt yr$^{-1}$ |

**Table 1.** List of ranges of values for the three temporal criteria for the top 90th percentile models, all CMIP5 models, and all CMIP6 models as well as the values and uncertainties for the reconstruction.

near the Davis Sea, and two opposite, weaker signals in Dronning Maud Land (Atlantic sector) and Adélie land (Pacific sector). This signal is reflective of the linear trend in SMB as seen in Fig. 2**B**. For a map of these Antarctic regions, see supplementary.

As a example of the comparison, one of the better scoring models for the EOF map criterion, CMCC CM, also shows a dipole between the Antarctic Peninsula and the Ross Sea region for the top mode as well as strong variance signal around the Antarctic Peninsula for mode 2 and a quadrupolar pattern for mode 3. However, even the better scoring models tend to overestimate the magnitude of the variance particularly around the coast even when they capture the general spatial patterns. CESM1 WACCM, one of the poorer performing models with regard to this metric, generally overestimates the variance everywhere in all three of the top modes. The top mode for this model reflects an East/West Antarctic SMB dipole and mode 2 shows a strong, unidirectional signal across the entire AIS, though mode 3 seems to reflect the same quadrupolar pattern as seen in the reconstruction.

Models that score above the 90th percentile make up the subset of best scoring models. Eight models – GISS E2 H CC, GISS E2 RCC, GISS E2 R, MPI ESM LR, MPI ESM MR, and MPI ESM P from CMIP5 and CESM FV2 and MPI ESM LR from CMIP6 – comprise this top 90th percentile. MPI ESM P GISS E2 R from CMIP5 and CESM2 FV2 do not have the requisite future projection data for this analysis. The poorest performing models include BNU ESM, CESM FASTCHEM, and FIO ESM. The mean model score is 4.36 for CMIP5 and 5.77 for CMIP6. CMIP5 and CMIP6 scores were normalized together such that all scores are on the same scale and are directly comparable.

With this subset of the eight best performing models, we then refined future projections of AIS SMB in terms of mean value, trend, and variability. Comparing the difference in SMB projections between RCPs/SSPs allows us a look into the potential sea level changes caused by different amounts of warming.

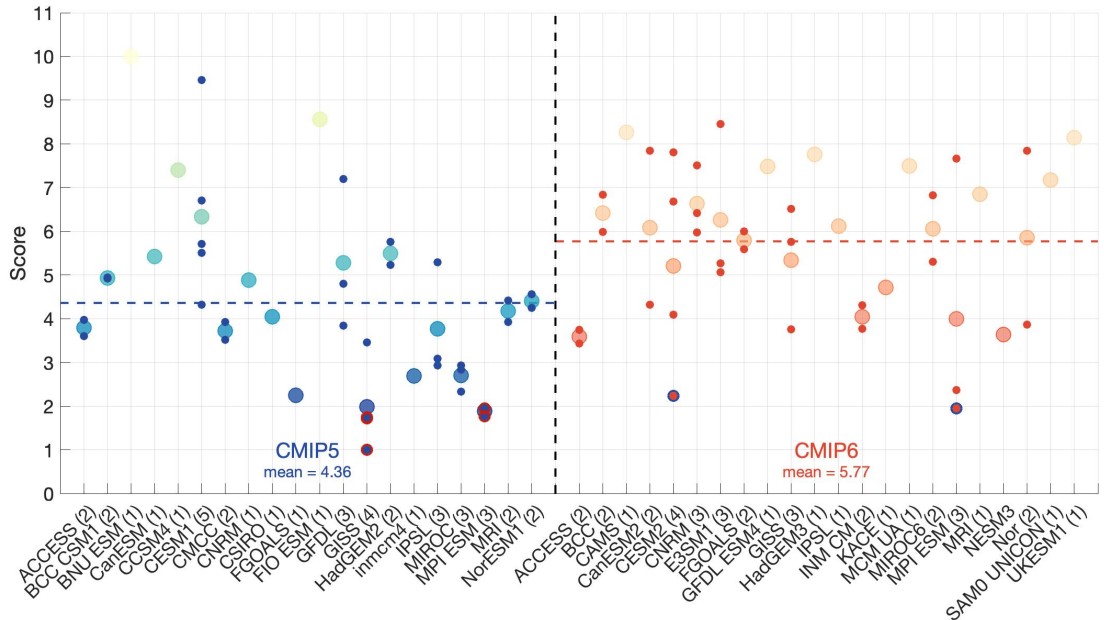

**Figure 7.** The scores for all CMIP5 and CMIP6 models. The large dots show the average score for all model groupings. Models are grouped by similar model physics and have in parenthesis the number of models in the grouping after the name. Each model grouping has all model scores plotted as small blue/red dots for CMIP5/6 with the model average plotted in the larger dots. Models that have no like models are followed by a one in parenthesis and only have a larger dot. The eight best scoring models (above the 90th percentile) are denoted with red outlines if they are among the CMIP5 suite of models – GISS E2 H CC, GISS E2 R CC, GISS E2 R, MPI ESM LR, MPI ESM MR, and MPI ESM P – or with blue outlines if they are among the CMIP6 suite of models – CESM FV2 and MPI ESM2 LR. Note that the overall scores for two of the GISS models and three of the MPI models in CMIP5 are almost exactly equal so outlines overlap almost completely.

As stated earlier, both mean value and trend of AIS SMB have significant implications for future projections of sea level change. The spatially integrated AIS SMB (i.e. SMB mean value) has been increasing from 1850-2000 (Fig. 3) and is projected to continue to increase for the following hundred years to 2100 in all three warming scenarios (Fig. 8).

From 2070-2100, spatially integrated AIS SMB is projected to be $2294 \pm 570$ Gt yr$^{-1}$ for RCP2.6, $2371 \pm 581$ Gt yr$^{-1}$ for RCP4.5, and $2358 \pm 663$ Gt yr$^{-1}$ for RCP8.5 for all CMIP5 models where the associated uncertainties are 1-$\sigma$ of all models 280  between 2070-2100 (for a list of projected SMB and related variable values for all models and the best scoring models across the RCPs, see supplementary). For the same time period in CMIP6, AIS SMB is projected to be $2249 \pm 392$ Gt yr$^{-1}$ for SSP1-2.6, $2305 \pm 387$ Gt yr$^{-1}$ for SSP2-4.5, and $2418 \pm 374$ Gt yr$^{-1}$ for SSP5-8.5. The subset of best scoring models have lower projections and smaller spread at $2274 \pm 282$ Gt yr$^{-1}$ for RCP2.6, $2358 \pm 286$ Gt yr$^{-1}$ for RCP4.5, and $2495 \pm 291$ Gt yr$^{-1}$ for RCP8.5 for CMIP5 between 2070-2100. For CMIP6 over the same period, the best scoring model, MRI ESM2, 285  projects AIS SMB to be even lower at 2073 Gt yr$^{-1}$ for SSP1-2.6, 2096 Gt yr$^{-1}$ for SSP2-4.5, and 2154 Gt yr$^{-1}$ for SSP5-8.5. The ranges of the best scoring models reduced the spread by 79%, 79%, and 74% for RCPs 2.6, 4.5, and 8.5, respectively. The

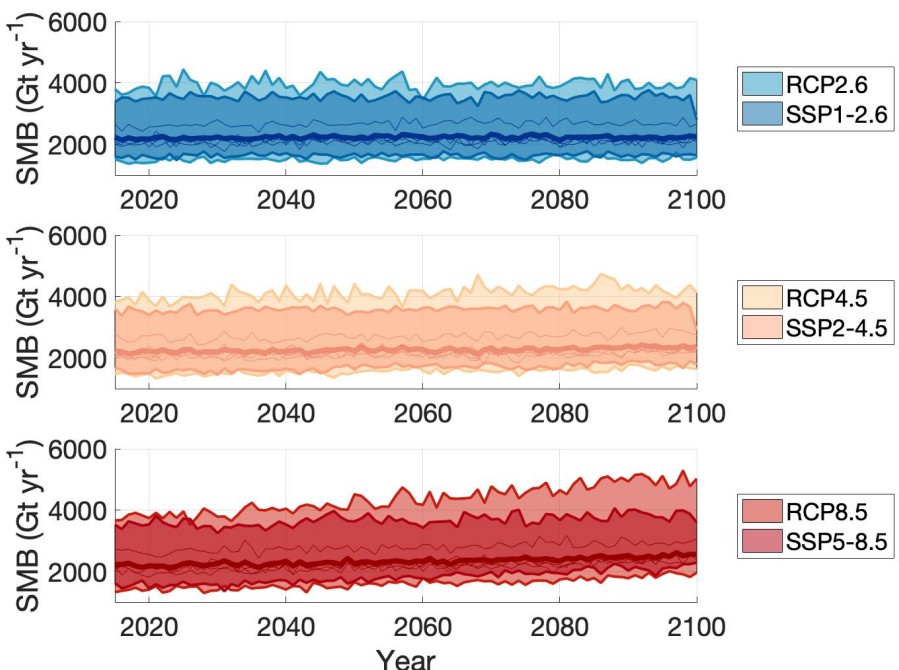

**Figure 8.** Time series for all CMIP5 (lighter colors) and CMIP6 models (darker colors) and best scoring models (skinny lines) and the best scoring models' average (thick lines) for **A** RCP2.6/SSP1-2.6 (blue), **B** RCP4.5/SSP2-4.5 (yellow), and **C** RCP8.5/SSP5-8.5 (red).

mean value of modeled SMB increases with increasing warming scenarios in all CMIP5 and CMIP6 models, as well as in the subset of the best scoring models. Similarly to the mean value increasing with increasing warming, the projected SMB trend also increases with increased warming (Fig. 9).

For the entirety of the 21st century, 2000-2100, most CMIP5 and CMIP6 climate models project positive SMB trends in all forcing scenarios (Fig. 9). For RCP2.6, all CMIP5 models project a mean trend of $0.9 \pm 1.2$ Gt yr$^{-2}$. For RCPs 4.5 and 8.5, the means trends are $2.5 \pm 1.6$ Gt yr$^{-2}$ and $6.0 \pm 3.2$ Gt yr$^{-2}$, respectively. In CMIP6, the mean trends are $1.3 \pm 1.3$ Gt yr$^{-2}$, $2.5 \pm 1.9$ Gt yr$^{-2}$, and $5.0 \pm 2.9$ Gt yr$^{-2}$ for SSP1-2.6, 2-4.5, and 5-8.5, respectively.

The best scoring CMIP5 models have trends of $1.2 \pm 1.0$ Gt yr$^{-2}$, $1.9 \pm 0.7$ Gt yr$^{-2}$, and $3.8 \pm 0.8$ yr$^{-2}$ for RCPs 2.6,
4.5, and 8.5, respectively. The best scoring CMIP6 model has trends of 0.5 Gt yr$^{-2}$, 2.0 Gt yr$^{-2}$, and 3.8 Gt yr$^{-2}$, for SSPs 1-2.6, 2-4.5, and 5-8.5, respectively. For RCPs 2.6 and 4.5 and SSPs 1-2.6 and 2-4.5, the best scoring model trend projections lie close to or within the interquartile range for all CMIP5 and CMIP6 models. As the warming scenarios strengthen, the five of the eight best scoring models projected into the future move closer to the lower end of the overall interquartile ranges in trend. Some of the differences in these concentration pathways can be described by the modeled SMB sensitivity to different
atmospheric $CO_2$ emission scenarios.

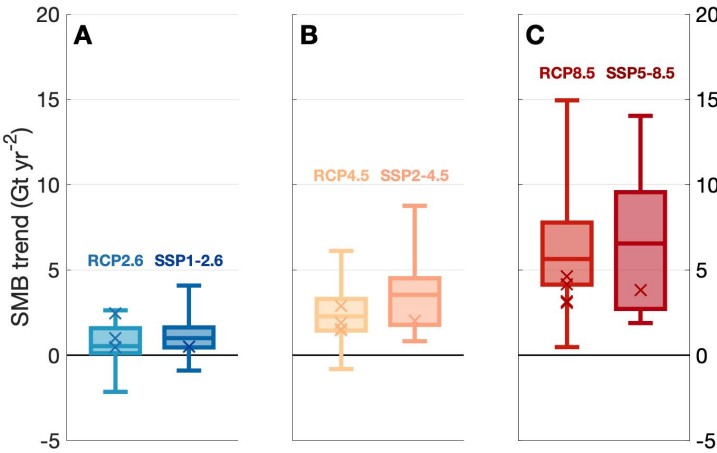

**Figure 9.** Box plots of the linear trend in spatially integrated AIS SMB from 2050-2100 for **A** RCP2.6/SSP1-2.6 (blue), **B** RCP4.5/SSP2-4.5 (yellow), and **C** RCP8.5/SSP5-8.5 (red). The five darker x's denote the five models – GISS E2 H CC, GISS E2 R CC, MPI ESM LR, and MPI ESM MR from CMIP5 and MRI ESM2 from CMIP6 – among the eight best scoring models with the appropriate and necessary information for direct comparison of future projections.

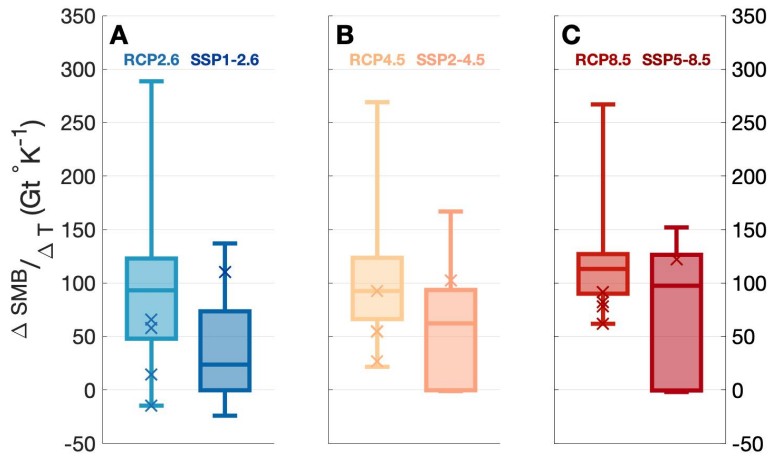

**Figure 10.** Box plots of all CMIP5 models' projected SMB sensitivity to temperature changes ($\Delta$SMB/$\Delta$T) for **A** RCP2.6/SSP1-2.6, **B** RCP4.5/SSP2-4.5, and **C** RCP8.5/SSP5-8.5. The five darker x's denote the five models – GISS E2 H CC, GISS E2 R CC, MPI ESM LR, and MPI ESM MR from CMIP5 and MRI ESM2 from CMIP6 – among the eight best scoring models with the appropriate and necessary information for direct comparison of future projections.

Box plots of modeled SMB sensitivity to changes in temperature (i.e. how much SMB will change per degree of near-surface atmospheric warming) are shown in Fig. 10. The projected sensitivity means for RCPs 2.6, 4.5, and 8.5 are $95 \pm 70$ Gt $^{\circ}$K$^{-1}$, $102 \pm 52$ Gt $^{\circ}$K$^{-1}$, and $120 \pm 46$ Gt $^{\circ}$K$^{-1}$, respectively. The four best scoring CMIP5 models are below the median for each forcing scenario with many of them below the lower limit of the interquartile ranges. The sensitivity in CMIP6 models SSPs 1-2.6, 2-4.5, and 5-8.5 are lower than those of CMIP5 at $39 \pm 49$ Gt $^{\circ}$K$^{-1}$, $59 \pm 53$ Gt $^{\circ}$K$^{-1}$, and $82 \pm 60$ Gt $^{\circ}$K$^{-1}$, respectively. The best scoring CMIP6 model is close to or above the upper limit of the interquartile range in sensitivity for each forcing scenario. CMIP5 shows a greater range in sensitivity for all three forcing scenarios as well as being generally more sensitive across all CMIP5 and CMIP6 models.

These sensitivity results are not statistically significantly different across forcing scenarios, however, indicating no significant more-than-linear SMB increase in enhanced warming scenarios. Table 2 displays ranges for SMB mean value, SMB trend, SMB sensitivity, and temperature changes for all models and the best scoring models for the different forcing scenarios.

# 5 Discussion

## 5.1 EOF Analysis

The differences in modes of variability in the EOF maps likely point to differences in atmospheric conditions that force AIS SMB. Mode 1 of the reconstruction EOF shows a dipolar pattern across the Antarctic Peninsula and Ross Ice Shelf region of West Antarctica. This dipole corresponds to variability in precipitation generated by variations in the track and strength of the Amundsen Sea Low. The Amundsen Sea Low, a dominant synoptic phenomenon that drives a significant amount of the circulation variability in West Antarctica and on the Antarctic Peninsula (Turner et al., 2013), is marked by high precipitation around the coast of the Antarctic Peninsula (Grieger et al., 2016). Changes in the Amundsen Sea Low synoptic pattern, then, represent the dominant cause of variability in the reconstruction SMB. The depth of the ASL is strongly influenced by the phase of the Southern annular mode (SAM) with positive (negative) mean sea level pressure anomalies when the SAM is negative (positive) (Turner et al., 2013).

Looking at mode 2, previous work by Hosking et al. (2013) and Turner et al. (2013) (among others) have shown that variability in the Amundsen Sea Low is responsible for high precipitation variability in West Antarctica and on the Antarctic Peninsula. Because this region dominates the overall AIS precipitation signal (as East Antarctica sees little snowfall by comparison), a variable Amundsen Sea Low signal, here, would explain the EOF pattern reflected in mode 2 of the reconstruction. Additional work highlighted in the supplementary material indicates that variability in sea level pressure in the Amundsen Sea region may be playing a large role in the AIS SMB spatial variability patterns.

| | | CMIP5 | | | | | |
| --- | --- | --- | --- | --- | --- | --- | --- |
| | | RCP2.6 | | RCP4.5 | | RCP8.5 | |
| | Years | All | Best | All | Best | All | Best |
| SMB (Gt yr$^{-1}$) | 2070-2100 | $2294 \pm 570$ | $2274 \pm 282$ | $2371 \pm 581$ | $2358 \pm 286$ | $2630 \pm 663$ | $2495 \pm 291$ |
| $\frac{\Delta \mathrm{SMB}}{\Delta t}$ (Gt yr$^{-2}$) | 2000-2100 | $0.9 \pm 1.2$ | $1.3 \pm 1.0$ | $2.5 \pm 1.6$ | $1.9 \pm 0.7$ | $6.0 \pm 3.2$ | $3.8 \pm 0.8$ |
| $\frac{\Delta \mathrm{SMB}}{\Delta T}$ (Gt yr$^{-1}$ °C$^{-1}$) | 2000-2100 | $95 \pm 70$ | $31 \pm 38$ | $102 \pm 53$ | $57 \pm 27$ | $120 \pm 46$ | $78 \pm 12$ |
| $\Delta T$ (°C 100yr$^{-1}$) | 2000-2100 | $0.8 \pm 0.8$ | $0.6 \pm 0.6$ | $1.2 \pm 1.0$ | $1.0 \pm 1.2$ | $4.8 \pm 1.0$ | $3.6 \pm 0.2$ |

| | | CMIP6 | | | | | |
| --- | --- | --- | --- | --- | --- | --- | --- |
| | | SSP1-2.6 | | SSP2-4.5 | | SSP5-8.5 | |
| | Years | All | Best* | All | Best* | All | Best* |
| SMB (Gt yr$^{-1}$) | 2070-2100 | $2249 \pm 392$ | 2073 | $2305 \pm 387$ | 2096 | $2418 \pm 374$ | 2154 |
| $\frac{\Delta \mathrm{SMB}}{\Delta t}$ (Gt yr$^{-2}$) | 2000-2100 | $1.3 \pm 1.3$ | 0.5 | $2.5 \pm 1.9$ | 2.0 | $5.0 \pm 2.9$ | 3.8 |
| $\frac{\Delta \mathrm{SMB}}{\Delta T}$ (Gt yr$^{-1}$ °C$^{-1}$) | 2000-2100 | $39 \pm 49$ | 110 | $59 \pm 53$ | 102 | $82 \pm 60$ | 122 |
| $\Delta T$ (°C 100yr$^{-1}$) | 2000-2100 | $0.9 \pm 0.8$ | 1.2 | $2.2 \pm 0.8$ | 3.0 | $5.2 \pm 1.6$ | 5.7 |

**Table 2.** *There is only one best scoring model with data for the future forcing scenarios so no uncertainty is provided. Projected values for SMB, SMB trend, SMB temperature sensitivity, and change in 21st century temperature for all CMIP5 and CMIP6 models compared to the best scoring CMIP5 models for RCP2.6, RCP4.5, and RCP8.5 and best scoring CMIP6 models for SSP1-2.6, SSP2-4.5, and SSP5-8.5.

## 5.2 Impact of internal variability on model scoring

Our study uses the full ensemble of available CMIP5 and CMIP6 models. However, we only select a single member of each model (since some models have only one ensemble member available), which potentially leads to under-sampling of internal variability in the scoring. To analyze the effect of natural variability on final scoring, we use the Large Ensemble of the Community Earth System Model (CESM-LENS, (Kay et al., 2015)). Because of its large number of ensemble members, the CESM-LENS experiment is useful for quantifying the role of internal variability. Only 35 of the original 40 ensemble members contain the necessary information for assessing AIS SMB. Figure 4 in Supplementary shows the final scores of the five CESM simulations that are included in the CMIP5 suite of models as well as the final scores of the CESM-LENS experiment. The final scores for the CESM-LENS model runs are calculated the same way for all model criteria except for AIS-integrated trend. Because these runs only differ after 1920, we only use the third time slice (1950-2000) to assess the quality of trend reproduction. The final scores of the five CMIP5 CESM model runs range from 3.99 to 9.74 while the final scores of the 35 CESM-LENS runs range from 1.32 to 5.96. Given that the scores range by 5.74 and 4.65 for the CMIP5 CESM runs and the CESM-LENS runs, respectively, it is reasonable to conclude that internal variability plays as significant a role in determining final score as do model parameterizations. This spread in score is due, in large part, to the spread among the ensemble members in spatial variability (EOF) patterns.

A major caveat of this finding, however, is that the CESM-LENS runs and the reconstruction only overlap from 1920-2000. This will likely most significantly impact the assessment of the trend and EOF analyses.

That said, this analysis highlights that internal variability plays a significant role in our AIS SMB assessment. Some models within the CMIP5 and CMIP6 frameworks, such as CESM1-CAM5, have many ensemble members. However, not all models – and even not all model versions – have multiple ensemble members. As such, performing a direct comparison of the models

using the ensemble mean would not necessarily yield an accurate result as models with more ensemble members would have their final score shifted significantly while the same is not true for models with a single ensemble member. For considering using GCMs for AIS SMB analysis, then, we strongly suggest taking into account the fact that internal variability could be playing a strong role in some models final score, and that the number of ensemble members available should be considered along with the final score.

## 5.3    Impact of horizontal resolution on scoring

As the CMIP5 and CMIP6 models vary widely in horizontal resolution, from about $0.75° \times 0.75°$ to $3° \times 3°$ (Tables 1-4 in Supplementary), we can assess the impact of resolution on individual and final model scoring. Figure 5 in Supplementary shows a scatter plot of resolution versus total score. Resolution, here, is the latitudinal resolution multiplied by the longitudinal resolution such that a model with latitude/longitude resolutions $0.9375°/1.25°$ would have a resolution of $1.1719°$. A linear

regression yields a correlation of R = -0.40 with 95% confidence intervals of -0.62 and -0.17. From this, there is a small, though statistically significant negative correlation between resolution and total model score, signaling that, perhaps contrary to intuition, lower-resolution models score equally well, if not better, than higher resolution models. When comparing total scores from the same model run at different resolutions, we find a consistent result: the relative high-resolution CESM CAM5, IPSL CM5A MR, MPI ESM MR, CESM2, CESM2 WACCM, and MPI ESM2 HR all perform worse than their coarser resolution

counterparts – CESM CAM5 FV2, IPSL CM5A LR, MPI ESM LR, CESM2 FV2, CESM2 WACCM FV2, and MPI ESM2 LR.

## 5.4    Caveats

The major limitations of this work stem largely from the subjective selection of scoring criteria. While each model is scored based on the same criteria, each criterion is chosen specifically to gauge model performance for capturing AIS SMB. As such,

these criteria may be ill suited for looking at other variables and, thus, other metrics could yield very different results. Another caveat of this work is that we are only capable of analyzing the CMIP6 models that have been released. As this analysis and the release of CMIP6 are concurrent, this limits the number of models we can reasonably analyze due to time constraints. Additional CMIP6 models may have different results and may skew the comparison between CMIP5 and CMIP6 significantly. Similarly, due to the small number of CMIP6 models released at this point, using statistical analyses becomes moot as the top

90% of models constitutes the single, best scoring model. One final major caveat with this work is the relatively narrow scope of just looking at AIS SMB. Because we refined our criteria at the outset of our experiment to solely reflect model performance with regard to capturing SMB and didn't include outside factors like synoptic weather patterns, sea ice or sea surface conditions

(Krinner et al. (2014); Kittel et al. (2018)), there are potentially some wider model biases that we are missing that could affect SMB projections. In our analysis, we make the significant assumption that the past ability to capture SMB correlates to higher skill in projecting AIS SMB into the future. However, model biases in some of the larger physical drivers – and how those biases change into the future – will significantly impact future AIS SMB trajectory.

Another significant caveat of this work is the use of single ensemble members. For this work, we use the first ensemble member for each model. This choice was made as the various model members of CMIP5 and CMIP6 vary widely in the number of ensemble members available – ranging from 1 to 50 – so using only a single ensemble members helps account for this large disparity between the models. However, in looking at the CESM-LENS experiment – which has 35 ensemble members – it is clear that there can be a large spread caused solely by internal variability. The spread in final score among the CESM-LENS ensemble members is 4.65 which is largely generated by the difference in EOF maps meaning that the precise realization of atmospheric conditions in the models is incredibly significant in how the model, in turn, represents AIS SMB.

## 6  Conclusions

In this paper, we tested the ability of the suite of models in CMIP5 to capture SMB reconstructed from ice cores and reanalysis products by scoring them using a series of criteria: AIS-integrated mean value, trend, and variability, as well as the spatial variability patterns. This scoring system is designed as a guide for choosing what GCMs to focus on studying for future SMB projections. Using this scoring system, we found that the top 90th percentile models were GISS E2 H CC, GISS E2 R CC, GISS E2 R, MPI ESM LR, MPI ESM MR, and MPI ESM P of CMIP5 and CESM FV2 and MPI ESM2 LR of CMIP6. A similar study in Agosta et al. (2015) found ACCESS1-3, ACCESS1-0, CESM BGC, CESM CAM5, NorESM1-M, and EC-Earth to most accurately capture AIS sea level pressure, 850 hPa air temperature, precipitable water, and ocean conditions – all of which impact AIS SMB to varying degrees. They focused their investigation into more atmospheric and oceanic dynamics (sea ice extent, sea surface temperature, sea surface pressure, precipitable water, 850 hPa temperature) and were comparing models directly to a reanalysis product. Barthel et al. (2019), another study with a similar goal of analyzing SMB performance among GCMs selected CCSM4, MIROC ESM CHEM, and NorESM1-M as their top three performing models for Antarctica. They ruled out both the GISS and MPI modeling groups due to their initial selection criteria and were also looking more at the impacts thermodynamical processes on SMB.

Our SMB mean value estimates are comparable to Agosta et al. (2019), who found a mean SMB value of roughly $2100 \pm 100$ Gt yr$^{-1}$ for the grounded AIS using ERA-Interim products. The SMB trends are also in line with Medley and Thomas (2019) over the 20th century. Unlike previous studies, we use a reconstructed data set based on ice core reanalysis, not RCMs. Also of note is the fact that this data set and the GCMs we use for comparison allow us to investigate much longer time periods (150 years), enhancing the robustness of long-term AIS SMB trends. Using this reconstruction, we are able to refine estimates of SMB mean value and SMB trend by the end of the 21st century using CMIP5 by assigning scores to the models and creating a subset of the most accurate models historically. Also unlike previous studies, we analyze both CMIP5 and the early models

of CMIP6 together allowing for direct comparison between the two suites of models. The scores for all CMIP5 models are, on average, better than the average score of the currently released CMIP6 models.

All scores are equally weighted to avoid issues with coincidental good or bad performance. Having a spread of criteria against which we score the models limits the possibility that models are recreating one aspect well for the wrong reasons. This scoring method does well in determining simple and consistent criteria to score the accuracy of modeled SMB. In contrast, it struggles to recognize any difference in the importance of individual criteria as they are all weighted equally and also only reflects a few, simple scoring metrics. The criteria were chosen such that they all carry equal weight which we justify by arguing that not meeting any one of the criteria to within a reasonable degree would significantly impact future SMB estimates.

Of the top eight scoring models, six were from CMIP5 and two from CMIP6. Using the top six best scoring models from CMIP5, four of which we were able to project out to 2100 under three different RCPs, we refined future SMB predictions to $2274 \pm 282$ Gt yr$^{-1}$ for RCP2.6, $2358 \pm 286$ Gt yr$^{-1}$ for RCP4.5, and $2495 \pm 291$ Gt yr$^{-1}$ for RCP8.5. Of the two best scoring CMIP6 models, only one (MRI ESM2) had data for the comparable future SSP forcing scenarios. For the 1-2.6 and 2-4.5 scenarios, MRI ESM2 is within the standard deviation of the CMIP5 models (albeit at the very low end). For the SSP5-8.5 scenario, MRI ESM2 is about 50 Gt yr$^{-1}$ less than the lower limit of the mean $\pm$ the standard deviation of the CMIP5 models. Our result of these best scoring models projecting AIS SMB at the lower end of the overall CMIP5 interquartile range in trend is in contrast to Palerme et al. (2017) who found that, especially considering RCPs 2.6 and 4.5, the CMIP5 models that best captured snowfall change rates tended to predict higher snowfall rates into the 21st century. The best scoring CMIP6 model similarly tends to fall at the lower end of the overall interquartile range. Additionally, model trends were refined to 0.47 to 2.45 Gt yr$^{-2}$ for RCP2.6, 1.44 to 2.88 Gt yr$^{-2}$ for RCP4.5, and 3.06 to 4.63 Gt yr$^{-2}$ for RCP8.5. MRI ESM2, the best scoring CMIP6 model, showed trends of 0.5 Gt yr$^{-2}$, 2.0 Gt yr$^{-2}$, and 3.8 Gt yr$^{-2}$ for SSPs 1-2.6, 2-4.5, and 5-8.5, respectively. Comparing the projected change in SMB per degree warming between the emission scenarios gives mean sensitivities of $31 \pm 38$ Gt $^{\circ}$K$^{-1}$, $57 \pm 27$ Gt $^{\circ}$K$^{-1}$, and $78 \pm 12$ Gt $^{\circ}$K$^{-1}$ for RCPs 2.6, 4.5, and 8.5, respectively, for the best scoring models. The best scoring CMIP6 model had sensitivities that were generally higher than the best scoring CMIP5 models at: 110 Gt $^{\circ}$K$^{-1}$, 102 Gt $^{\circ}$K$^{-1}$, and 122 Gt $^{\circ}$K$^{-1}$ for SSP1-2.6, SSP2-4.5, and SSP5-8.5, respective. (For a list of all values for CMIP5 and CMIP6 models, see Table 2.) However, the sensitivity results from CMIP5 are not statistically significantly different from one another across forcing scenarios and indicate that there is no difference in the sensitivity response to changes in temperature between the three forcing scenarios. Given that the best performing models show lower AIS-integrated SMB values and trends compared to the entire CMIP5 spread indicates less sea level rise mitigation from increasing SMB than is implied by looking at all CMIP5 models.

Some of the major caveats of this work are the subjective selection of scoring criteria which dictate the assessment of best scoring models as well as the use of single-ensemble members for model analysis which may lead to an undersampling of internal variability.

# Appendix A

## A1

*Author contributions.* T. G. and J. T. M. L. conceptualized and initiated this work. T. G. performed the analysis, discussed the results with J. T. M. L., and wrote the paper. B. M. provided the reconstructions and guidance on using and interpreting them. All authors reviewed the paper before submission.

*Competing interests.* The authors declare no competing interests.

*Acknowledgements.* T. G. and J. T. M. L. acknowledge support from the National Aeronatics and Space Administration (NASA), Grant 80NSSC17K0565 (NASA Sea Level Team 2017–2020).

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

# Scoring Antarctic surface mass balance in climate models to refine future projections

Tessa Gorte[1], Jan T. M. Lenaerts[1], and Brooke Medley[2]

[1]Department of Atmospheric and Oceanic Sciences, University of Colorado Boulder
[2]Cryospheric Sciences Laboratory, National Aeronautics and Space Administration's Goddard Space Flight Center

## 1   Empirical Orthogonal Functions

The top three modes of AIS SMB variability in the reconstruction are the only three modes with the percent of variance explained above 10% (Fig. 1). In total, these top three modes explain about 77% of the total variance in AIS SMB in the reconstruction.

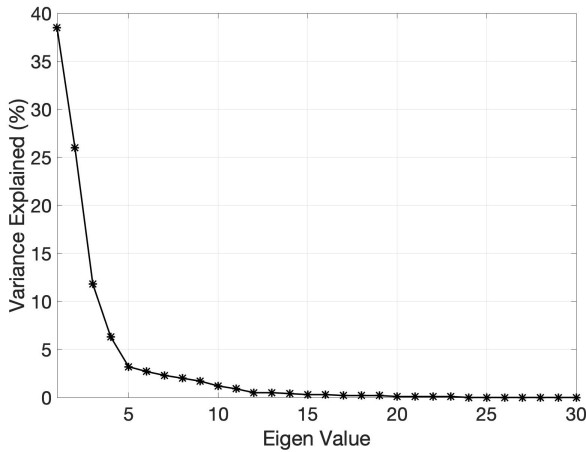

**Figure 1.** The top 30 Eigen values out of 200 total for SMB in the reconstruction. The top three Eigenvalues explain 76.3% of the total SMB variance.

To gain insight into what atmospheric conditions may lead to the dominant modes of SMB spatial variability, we performed the same EOF analysis on the reconstructed sea level pressure (Fig. 2). The top mode of atmospheric variability shows high variability around the Amundsen Sea region. Similarly, mode 2 also reflects strong variability in the Amundsen Sea region but with more zonal symmetry. The third mode of atmospheric variability represents a quadripolar pattern in variability about the 0-180° and 90°E/W longitude lines.

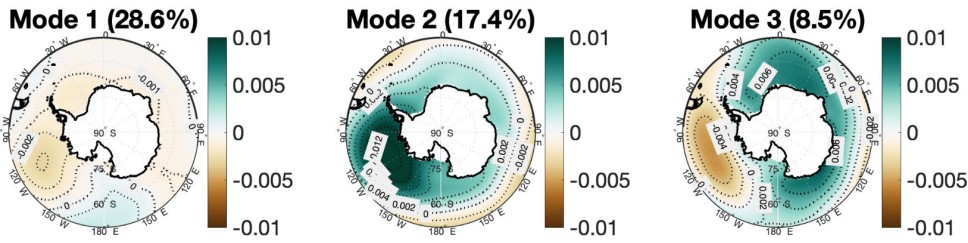

**Figure 2.** EOFs of the top 3 modes of the reconstruction for sea level pressure.

## 2  Sample Size Monte Carlo

The average final score for CMIP5 is 3.7 and the average final score for CMIP5 is 5.6. To determine if this difference is generated by the smaller CMIP6 sample size, we performed a Monte Carlo-type simulation. Randomly selecting 22 of the 41 total CMIP5 model scores 10,000 times, we then tested whether those 10,000 selections were statistically different from all 41 using a two-sided t-test. The t-test generates results of 0 if we cannot reject the null hypothesis that the two samples are different at the 95% confidence level or 1 if we can. Averaging the t-test over all 10,000 selections yields a 0.042% chance that we can reject the null hypothesis. From this, we determine that we cannot reasonably reject the null hypothesis that these two scores are statistically different at the 95% confidence level. This means that 22 models is representative of the total CMIP5 suite of models from which we hypothesize that the same can be said for the current 22 models being representative of the full CMIP6 suite of models in terms of average final score.

## 3  Future temperature and SMB trends

To assess how much of the model sensitivity to forcing scenario is attributable to spread in $\Delta T$ versus spread in $\frac{\Delta \text{SMB}}{\Delta T}$, we compared the relative spreads of each. For RCPs 2.6, 4.5, and 8.5, respectively, $\frac{\Delta \text{SMB}}{\Delta T}$ ranged between -116% to + 305%, 21% to 264%, and 52% to 223% about their respective means. By comparison, $\Delta T$ ranged between 56% to 156%, 30% to 141%, and 45% to 135% about their relative means for RCPs 2.6, 4.5, and 8.5, respectively. In short, $\Delta T$ ranged about 100% about the mean in each scenario while $\frac{\Delta \text{SMB}}{\Delta T}$ ranged about 200% to 300% about the mean depending on scenario. With that, we conclude that much of the variation in $\frac{\Delta \text{SMB}}{\Delta T}$ between models stems from differences in how the models react to different forcing scenarios rather than owing to large spread in modeled temperature change over the 21st century.

## 4  Impact of Internal Variability in Model Scoring: CESM Large Ensemble

Internal variability – the process by which model ensemble members deviate due to small changes in model initialization – potentially plays a large role in overall model score. Figure 4 shows the scores of the 5 CESM members that appear in the CMIP5 suite as well as 35 ensemble members from the CESM-LENS experiment. The spread in CESM-LENS is comparable

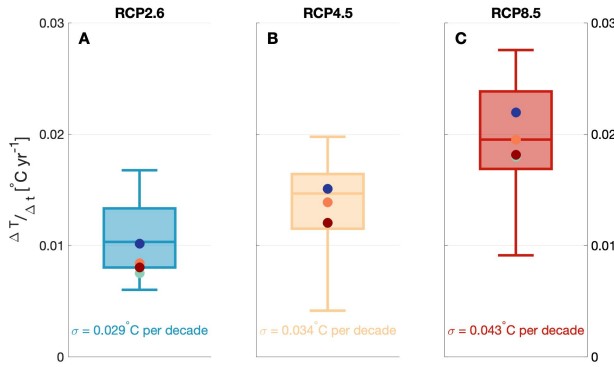

**Figure 3.** Box plots of the temperature trends in °C yr$^{-1}$ for **A** RCP2.6 (blue), **B** RCP4.5 (yellow), and **C** RCP8.5 (red). The four best scoring models are shown as colored circles: GISS E2 H (dark blue), GISS E2 R (green), MPI ESM LR (coral), and MPI ESM MR (dark red).

to that of the CMIP5 CESM simulations. This spread is largely generated by variability in the EOF criteria. From the sizeable spread, we conclude that internal variability can potentially be as significant as specific parameterization choices within a single model. The spread in CESM-LENS is significantly smaller, though, than the spread across all CMIP5 models indicating that
35   model physics are the dominant factor in the reproduction of AIS SMB.

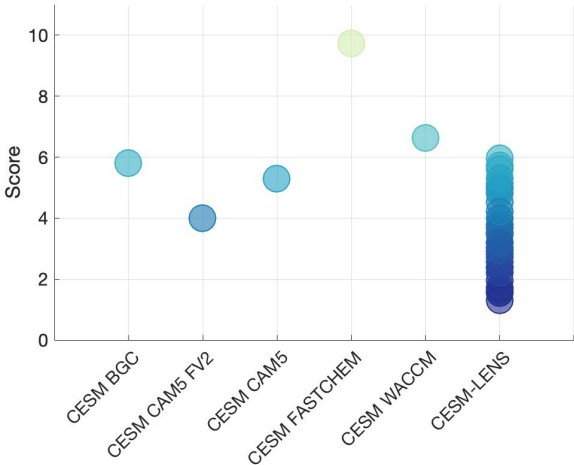

**Figure 4.** Final scores of the five CESM models from CMIP5 compared to the CEMS-LENS simulations.

## 4.1 Reconstruction Uncertainty and Intra-ensemble Variability

We calculated the reconstruction uncertainty used in the AIS-integrated criteria as

$$recon.uncertainty = \sqrt{(obs.uncertainty)^2 + (intra-ensemble\,variability)^2}. \tag{1}$$

Through our analysis of the CESM LENS experiment, we generated the intra-ensemble variability as one standard deviation of the raw values of AIS-integrated, temporally averaged mean value, AIS-integrated trend, and AIS-integrated variability (Fig. 5). With this intra-ensemble variability, we then calculated the reconstructed uncertainty used throughout the paper. The difference between the observational uncertainty and the reconstruction uncertainty for the three AIS-integrated criteria is small which aligns with our analysis that the EOF criteria generated the largest spread in score among the CESM LENS ensemble members.

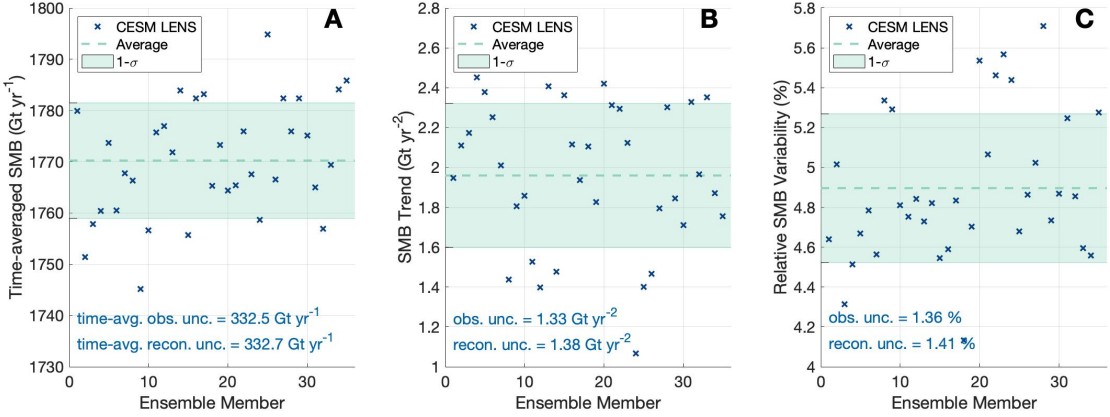

**Figure 5.** Scatter plots of CESM LENS ensemble member and their AIS-integrated, temporally averaged mean value **A**, AIS-integrated trend **B**, and AIS-integrated variability **C**. The dashed green line shows the ensemble average and the green shaded box denotes one standard deviation (1-$\sigma$).

## 5 Impact of Model Resolution in Model Scoring

To investigate the importance of model resolution for overall model score, we perform a linear regression analysis. Figure 6 shows all model resolutions plotted against their overall scores. Here, regression analysis shows a statistically significant, albeit relatively small, correlation coefficient of -0.4 with 95% confidence intervals of -0.62 and -0.17.

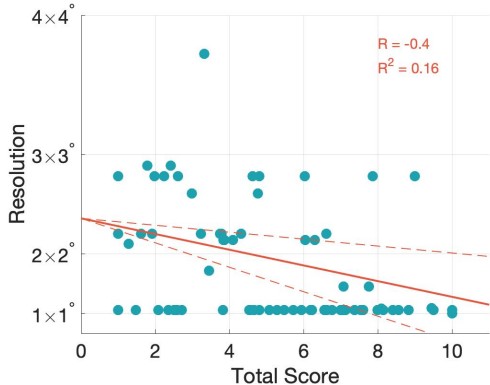

**Figure 6.** Scatter plot of resolution versus total score. A linear regression (orange line) yields a correlation of R = 0.45 with 95% confidence intervals (orange dashed lines) of 0.30 and 0.61.

## 6 Modeling Centers

Of the eight best scoring models, seven originate from two modeling centers: the Max Planck Institute fr Meteorologie (MPI) and Goddard Institute for Space Studies (GISS) from NASA. This strongly implies that model physics plays a significant role in the representation of AIS SMB. Another interpretation could be, though, that these models simply share the same biases and, thus, all are coincidentally favorably compared to the reconstruction. Here, we also look at a more diverse spread of modeling centers in two ways: 1) the top eight models that originate from unique modeling centers and 2) the top four modeling centers

(top 90th percentile) averaged across their members. Figures 8 and 7 show the best scoring eight models from unique modeling centers and best four scoring modeling centers on average, respectively. The former category consists of GISS R, MPI ESM LR, CESM2 FV2, FGOALS G2, MIROC ESM, INM CM4, IPSL CM5A MR, and ACCESS ESM1-5 (Fig. 8). MPI ESM, GISS, FGOALS, and INM CM from CMIP5 constitute the latter category of best modeling centers on average (Fig. 7).

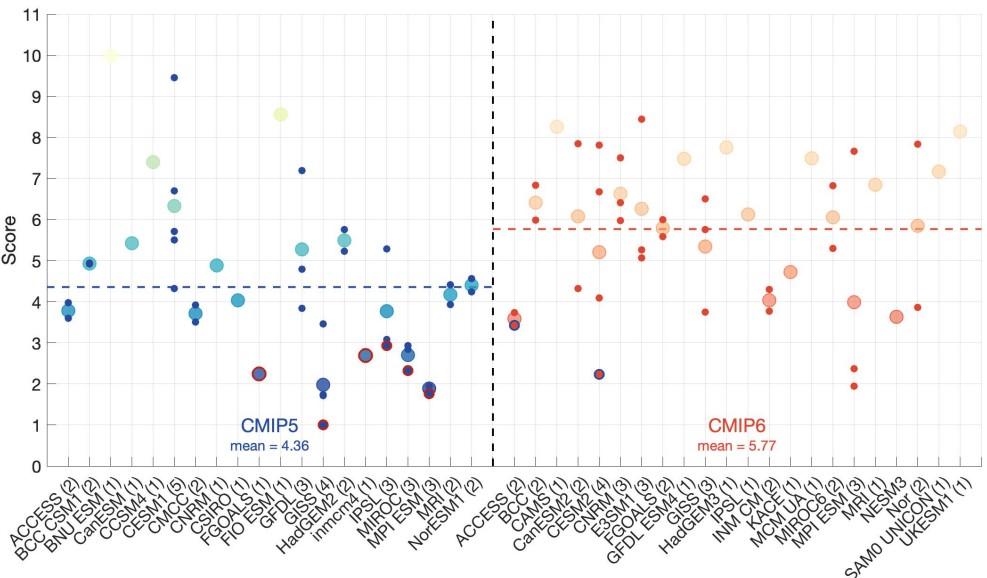

**Figure 7.**

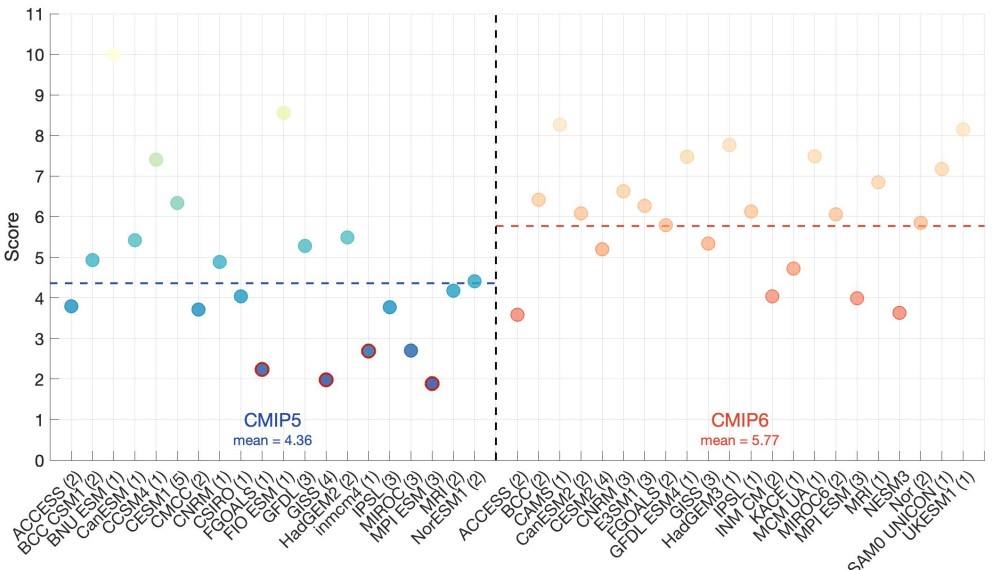

**Figure 8.**

# 7 Data Tables

This section includes tables with model resolutions and scores for all CMIP5 and CMIP6 models as well as a table with projected SMB and related variables for the various RCPs.

|    | Model | Resolution (°lat×°lon) | Total Score |
|----|-------|------------------------|-------------|
| 1  | ACCESS1-0 | 1.2414×1.875 | 3.80 |
| 2  | ACCESS1-3 | 1.2414×1.875 | 3.38 |
| 3  | BCC-CSM1.1 | 2.8125×2.8125 | 4.85 |
| 4  | BCC-CSM1.1-m | 2.8125×2.8125 | 4.99 |
| 5  | BNU ESM | 2.8125×2.8125 | 10 |
| 6  | CanESM2 | 2.8125×2.8125 | 5.24 |
| 7  | CCSM4 | 0.9375×1.25 | 7.40 |
| 8  | CESM1 BGC | 0.9375×1.25 | 5.81 |
| 9  | CESM1 CAM5 FV2 | 0.9375×1.25 | 3.99 |
| 10 | CESM1 CAM5 | 0.9375×1.25 | 5.30 |
| 11 | CESM1 FASTCHEM | 0.9375×1.25 | 9.74 |
| 12 | CESM1 WACCM | 0.9375×1.25 | 6.63 |
| 13 | CMCC CESM | 0.75×0.75 | 3.94 |
| 14 | CMCC CM | 0.75×0.75 | 3.50 |
| 15 | CNRM CM5 | 1.4063×1.4063 | 4.73 |
| 16 | CSIRO | 1.875×1.875 | 3.98 |
| 17 | FGOALS | 3×2.8125 | 2.07 |
| 18 | FIO ESM | 2.8125×2.8125 | 8.89 |
| 19 | GFDL CM3 | 2×2.5 | 3.83 |
| 20 | GFDL ESM2G | 2×2.5 | 4.48 |
| 21 | GFDL ESM2M | 2×2.5 | 7.11 |
| 22 | GISS E2 H CC | 2×2.5 | 1.72 |
| 23 | GISS E2 H | 2×2.5 | 3.55 |
| 24 | GISS E2 R CC | 2×2.5 | 1.60 |
| 25 | GISS E2 R | 2×2.5 | 1.00 |

**Table 1.** Model names, resolutions and final score for the first half of the CMIP5 suite of models.

|    | Model | Resolution (°lat×°lon) | Total Score |
|----|-------|------------------------|-------------|
| 26 | HadGEM2 CC | 1.2414×1.875 | 5.14 |
| 27 | HadGEM2 ES | 1.2414×1.875 | 5.56 |
| 28 | INMCM4 | 1.5×2 | 2.61 |
| 29 | IPSL CM5A LR | 1.875×3.75 | 3.13 |
| 30 | IPSL CM5A MR | 1.2587×2.5 | 2.88 |
| 31 | IPSL CM5B LR | 1.875×3.75 | 5.50 |
| 32 | MIROC ESM CHEM | 1.4063×1.4063 | 2.75 |
| 33 | MIROC ESM | 2.8125×2.8125 | 2.86 |
| 34 | MIROC5 | 2.8125×2.8125 | 2.19 |
| 35 | MPI ESM LR | 1.875×1.875 | 1.59 |
| 36 | MPI ESM MR | 1.067×1.067 | 1.76 |
| 37 | MPI ESM P | 1.875×1.875 | 1.76 |
| 38 | MRI CGCM3 | 1.125×1.125 | 3.92 |
| 39 | MRI ESM1 | 1.125×1.125 | 4.46 |
| 40 | NorESM1 M | 1.875×2.5 | 3.89 |
| 41 | NorESM1 ME | 1.875×2.5 | 4.30 |

65 **Table 2.** Model names, resolutions and final score for the second half of the CMIP5 suite of models.

|  | Model | Resolution (°lat×°lon) | Total Score |
|---|---|---|---|
| 1 | ACCESS CM2 | 1.25×1.875 | 3.66 |
| 2 | ACCESS ESM1-5 | 1.25×1.875 | 3.25 |
| 3 | BCC CSM2 MR | 1.125×1.125 | 7.17 |
| 4 | BCC ESM1 | 2.8125×2.8125 | 6.20 |
| 5 | CAMS CSM1 | 1.125×1.125 | 8.78 |
| 6 | CanESM5 | 2.8125×2.8125 | 8.25 |
| 7 | CanESM5-CanOE | 2.8125×2.8125 | 4.42 |
| 8 | CESM2 | 0.9375×1.25 | 8.04 |
| 9 | CESM2 FV2 | 1.9×2.5 | 2.08 |
| 10 | CESM2 WACCM | 0.9375×1.25 | 6.93 |
| 11 | CESM2 WACCM FV2 | 1.9×2.5 | 3.76 |
| 12 | CNRM CM6-1 | 1.4063×1.4063 | 6.63 |
| 13 | CNRM CM6-1-HR | 0.5×0.5 | 8.10 |
| 14 | CNRM ESM2 | 1.4063×1.4063 | 6.09 |
| 15 | E3SM1 | 1.0×1.0 | 8.79 |
| 16 | E3SM1-1 | 1.0×1.0 | 5.17 |
| 17 | E3SM1-1 ECA | 1.0×1.0 | 5.09 |
| 18 | FGOALS F3 L | 2.0×2.25 | 5.88 |
| 19 | FGOALS G3 | 2.0×2.25 | 5.72 |
| 20 | GFDL ESM4 | 1.0×1.25 | 7.71 |
| 21 | GISS E2 G | 2.0×2.5 | 5.81 |
| 22 | GISS E2 G CC | 2.0×2.5 | 3.55 |
| 23 | GISS E2 H | 2.0×2.5 | 6.78 |
| 24 | HadGEM3 GC3 | 1.25×1.875 | 7.97 |
| 25 | IPSL CM6A | 1.2587×2.5 | 6.30 |
| 26 | INM CM4-8 | 1.5×2.0 | 3.66 |
| 27 | INM CM5-0 | 1.5×2.0 | 4.04 |
| 28 | KACE1-0-G | 1.25×1.875 | 4.78 |
| 29 | MCM UA1 | 2.25×3.75 | 8.42 |
| 30 | MIROC6 | 2.8125×2.8125 | 7.14 |
| 31 | MIROC E2SL | 2.8125×2.8125 | 5.38 |

**Table 3.** Model names, resolutions and final score for the first half of CMIP6 suite of models.

| | Model | Resolution (°lat×°lon) | Total Score |
|---|---|---|---|
| 32 | MPI ESM1-2-HAM | 1.875×1.875 | 2.16 |
| 33 | MPI ESM1-2-HR | 0.935×0.935 | 8.04 |
| 34 | MPI ESM1-2-LR | 1.875×1.875 | 1.77 |
| 35 | MRI ESM2 | 1.125×1.125 | 7.16 |
| 36 | NESM3 | 1.875×1.875 | 3.45 |
| 37 | NorCPM1 | 1.875×2.5 | 8.00 |
| 38 | NorESM2-MM | 0.9375×1.25 | 3.80 |
| 39 | SAM0 UNICON | 0.9375×1.25 | 7.58 |
| 40 | UKESM1 | 0.9375×1.875 | 8.45 |

**Table 4.** Model names, resolutions and final score for the second half of CMIP6 suite of models.

## 8   AIS Map

Figure 9 shows the AIS with the names of locations specifically mentioned in the main text.

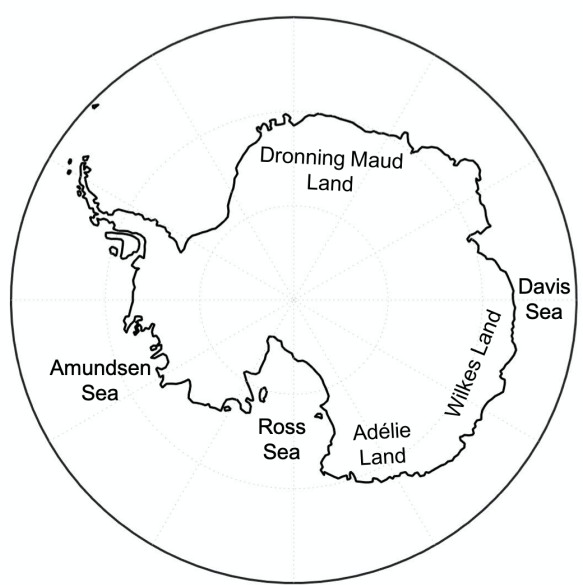

**Figure 9.** Map of the AIS.