# Peer review of "Scoring Antarctic surface mass balance in climate models to refine future projections"

_The Cryosphere, 2019_

## Referee Comment (RC1) · Anonymous Referee #1 · 14 Jan 2020

In this article, the authors use a reconstruction of the Antarctic surface mass balance, annually resolved, from ice cores, to evaluate CMIP5 and CMIP6 global climate models and to constrain projections based on their evaluation criteria.

I think this is an important subject that deserve to be published in TC, I found the paper generally well written with nicely shaped figures. However I think the paper need major revision before being published.

**Major**

*Methodology*

"To score the time series magnitude, we assigned a score, x, for how many x-times the reconstruction uncertainty was requiredÂăfor the entire time series to be within the

reconstruction uncertainty."

* I think you should reformulate this sentence in a more mathematical framework. What did you code? What is the minimum value of your score, 0 or 1?

* if I understand well, you did max(abs(Model - obs))/(reconstruction uncertainty)? So you scaled the maximum difference of model to obs with the reconstruction uncertainty? Why not using the RMSE scaled by the reconstruction uncertainty?

"involves finding what spatial SMB patterns explain the highest variance in the AIS-integrated SMB time."

* are you sure this what EOF do? Is is not the variance of space-time SMB variability?

* time series (typo?)

"To avoid manually sorting the top three modes of variability for all 53 models, we generated difference maps between each of the top three reconstructed modes and each of the top three modes for each model:"

* why do you do this only for the top 3 modes of each model and not e.g. the top 10?

"We then sorted the top modes of variability for each model based on smallest difference"

* what do you call "the smallest difference"? Do you average absolute differences over the map? Do you compute a RMSE?

"After compiling scores for all five of the aforementioned scoring criteria, we normalized each set of scores to be on a scale from one to ten to ensure that each criterion was equally weighted."

* So, if I understand well, you divide each criteria by the max of the criteria? This scaling is extremely sensitive to outliers. You should consider scaling by the interquantile range or by the standard deviation of each of you criteria.

"To refine the scope of what we predict for SMB in the future, we used a subsetÂăof models that had a final score in the top 10th percentile of CMIP5 and compared them to the entire scope of CMIP5"

* I am not sure it is a correct method. How much is your method sensitive to the number of models you keep? Why do you use this "10th percentile" criteria? I think that 4 models is too little to compute a robust statistic. Is it statistically correct to compare 4 members to 30 members? You should consider e.g. ensemble regression based on models' scores (Bracegirdle and Stephenson, 2012, doi: 10.1007/s00382-012-1330-3)

*Results*

Figure 1.
* when I see the spatial pattern of trends in 1B and 1C, I wonder why you use a criteria for SMB-integrated values instead of comparing spatial maps of trends? I think using spatial maps of trends would be more relevant.

"Looking at multiple time "slices" allows us to investigate if models capture the reconstructed SMB trends for the whole time series compared to more recent decades. Here, we looked at three time slices: the entire over-lapping time series from 1850-2000, the last century from 1900-2000, and the last 50 years from 1950-2000."
* I understand that simulating correctly the trends for 1950-2000 may be useful because it quantifies if the global climate models are able to simulate correctly the response to anthropogenic forcing. However I don't think that scoring the trends over the century is useful for your purpose. Your uncertainty on century-scale trends is very small and I wonder if it is not underestimated. It seems difficult to estimate century-scale internal variability from a 200 year reconstruction in fact.

"All CMIP5 and CMIP6 models overestimate SMB variability.ÂăThe CMIP5 and CMIP6 models range from overestimates of 144% to 261% and 151% to 217% of the reconstruction standard deviation, respectively"
* A strong warning here. I have doubts on the reliability of the reconstruction for interannual variability. How does the reconstruction interannual variability compare with the reanalyses variability for the common period? I suspect that the annual accumulation signal extracted from ice cores is dampened.

"This dipole corresponds to variability in precipitation generated by variations in the track and strength of the Amundsen Sea Low. The Amundsen Sea Low, which represents the pole of circulation variability in Antarctica (Turner et al., 2013), is marked by high precipitation around the coast of the Antarctic Peninsula (Grieger et al., 2016)."
* All this sentence is strange. It is more a discussion than a result. "The Amundsen Sea Low, which represents the pole of circulation variability in Antarctica"? What is a pole of circulation variability?

"The second mode of variability represents high variability in West Antarctica and the Antarctic Peninsula. This could be caused by the topography in these regions which can induce large amounts of snowfall."
* I am not sure that you interpret the EOFs correctly. The spatial pattern of an EOF associated to its time series explains to a certain amount of the space-time variability, but it does not mean that where the EOF spatial pattern is high there is a high variability.
* " This could be caused by the topography in these regions which can induce large amounts of snowfall." I don't understand why?

"By comparison, one of the better scoring models for the EOF map criterion, CMCC CM, also shows a dipole between the Antarctic Peninsula and the Ross Sea region for the top mode as well as strong variance signal around the Antarctic Peninsula for mode 2 and a quadrupolar pattern for mode 3."
* When looking at Fig. 5, EOF modes from the two climate models do not ressemble the reconstruction EOF modes, even for the best performing model (row B). Maybe showing the patterns with the same sign as for the reconstruction modes will help (multiply by -1 the climate model patterns). But still, they will remain very different. E.g. in row B there is no high spot near Davis for EOF 3, and there is a large dipole in WAIS. Are you sure of your computation? If yes, are you sure your analysis is relevant?

* What are the biases of the best scoring models for the large scale circulation fields (e.g. sea level pressure over Southern Ocean) over the last 40 years?

Fig 9 and associated text :
* The climate sensitivity for SMB must be shown in % K-1, because SMB varies exponentially with temperature. You should revise the end of section 4 with regard to climate sensitivities computed in % K-1.

* Given the issues on the scoring and the relevance of selecting four models, the new version of the manuscript might give different results.

**Minor**

"Integrated over the grounded Antarctic ice sheet (AIS), the blowing snow and runoff terms are negligibly small (Lenaerts et al., 2012a)."
* Drifting snow sublimation is still not well modeled and evaluated. You should reformulate, e.g. something like "we neglect blowing snow and runoff and estimate SMB as precipitation minus sublimation"

"Over longer time scales"
* Which ones?

"The strong regional variability suggests an important impact of variations in synoptic-scale patterns around the AIS (Fyke et al. (2017); Marshall et al. (2017))."
* It is known that synoptic scale patters drive the accumulation variability, reformulate, e.g. "Synoptic-scale variability induces a strong regional variability of the SMB"

"Additionally, as the atmosphere has been warming over large parts of the AIS and can hold more moisture per the Clausius-Clapeyron relation, SMB is expected to show an overall increase"
* Previdi and Polvani (2016, https://iopscience.iop.org/article/10.1088/1748-9326/11/9/094001) state that "the forced SMB increase due to global warming in recent decades is unlikely tobe detectable as a result of large natural SMB variability". Your sentence is unclear and potentially wrong for the last decades. Modify and add references.

"but many of those models tend to overestimate annual precipitation values due to poor representation of coastal topography"
* Are you sure it is because of the poor representation of coastal topography?

"This allows the atmospheric moisture to penetrate too far inland and leads to excessive precipitation on much of the grounded AIS, while underestimating precipitation nearby the coasts (Lenaerts et al. (2012b))."
* I did not read again this article, but it is about "Modeling drifting snow in Antarctica with a regional climate model: 1. Methods and model evaluation", so I am not sure it is the right paper to cite here? Do you have other references to show that resolution is the most important factor for modelling Antarctic precipitation?

"Barthel et al. (2019) investigated the Ice Sheet Model Intercomparison Project version 6 to determine a recommendation of which models to use for ice sheet model forcings based on best captured current Antarctic climate relative to observations and their ability to project certain metrics into the future"
* It's "Ice Sheet Model Intercomparison Project *for CMIP6*" and not "version 6" (in fact it's version 1).
* Barthel et al. (2019) evaluate the global climat models based on their ability to capture the large scale circulation around ice sheets compared to reanalyses . It is not "very similar" to your study because the "observation" they use is well evaluated (reanalyses large scale fields after 1979) and they don't use this criteria to constrain future projections.

"To improve upon model estimates, several groups have combined ice core data with models to create spatio-temporally robust SMB data sets (Monaghan et al. (2006), Thomas et al. (2017), Medley and Thomas (2019))."
* this sentence should be in the Method section

"In this work, we leverage the availability of that new avenue for climate model evaluation of AIS SMB, and compare the suite of CMIP5 and CMIP6 climate models to that

new SMB reconstruction."
* repetition of the sentence P2 L50-52, merge the two.

"they weighted each ice core spatially to generate the 200-year data set"
* give the period

"they calculated spatial sampling uncertainty is basedÂăon the RMSE"
* "they calculated spatial sampling uncertainty basedÂăon the RMSE"

"Global climate models tend to show higher skill at representing interannual variability compared to regional climate models (Medley and Thomas, 2019)."
* it is not what is said in Medley and Thomas, 2019. They say "Because of their aforementioned ability to reproduce the interannual variability[17], which strengthens the weighting scheme, we used *global atmospheric reanalyses* over regional climate models.". So this statement is for *reanalyses* compared to RCM only, and is based on [17] Medley, B. et al. Airborne-radar and ice-core observations of annual snow accumulation over Thwaites Glacier, West Antarctica confirm the spatiotemporal variability of global and regional atmospheric models. Geophys. Res. Lett. 40, 3649–3654 (2013).

"To get a comprehensive look at how well global climate models capture SMB, we compared the suite of CMIP5 models to the reconstruction."
* and CMIP6?

P4 L90-95
* I am not sure the detail of conversion of kg m-2 s-1 in Gt yr-1 is useful. Just saying that it is computed on the original GCM grid is enough.

P4 L99-100: remove parentheses

P4 L107: "the magnitude of the SMB time series"
* do you mean "the SMB mean value"? If yes it seems clearer for me to replace "magnitude" by "mean value" everywhere.

Interactive
comment

"To achieve this goal, we analyzed trends from 1850-2000, 1900-2000, and 1850-2000."
* typo
how do you combine the 3 periods?

"To score the time series variability, we detrended and normalized each time series to separate the SMB trend from its absolute magnitude using:"
* I don't understand "to separate the SMB trend from its absolute magnitude"

"To do so, we performed an empirical orthogonal function (EOF) analysis"
* on annual data over 1850-2005(?)

"By breaking this criterion down into two main factors, we were able to determine the models' abilities to accurately capture the modes of variability as well as how much variance each mode explained."
* what are the two main factors you are talking about?

P5 L169 "All four of best scoring models are captured within the reconstructed uncertainty for the entire 150 year time series."
* After reading further I understood that the best scoring models are for the combination of criteria. I think you should begin your result section by presenting the best scoring models (currently presented P10 and in the Figures' legends)

"We weighted all scores from the five scoring criteria equally on a scale from 1 to 10 with lower scores indicating better performance.ÂăThe final score, then, is the sum of all the individual scores, which is renormalized on a scale of 1 to 10 with lower scores still indicating better performance."
* repetition of P5 L141-143

" The reconstructed AIS SMB averaged from 1801-2000 shows higher SMB values around the coastal areas, particularly in the Antarctic Peninsula and West Antarctic regions (Fig. 1A)."

[Figure]

* This is really the most basic feature of Antarctic SMB, this sentence is not useful.

---

## Referee Comment (RC2) · Anonymous Referee #2 · 16 Jan 2020

This discussion paper assess surface mass balance (SMB) across CMIP5 and CMIP6 based on a specific set of metrics. A subset of four CMIP5 models is selected to refine 21st century Antarctic mean AIS SMB projections. They show weaker sensitivity and a smaller inter-model spread than the full ensemble. This is an important topic due to the implications for Antarctica's contribution to future global sea level change.

Unfortunately I do not feel that this discussion paper in its present form is suitable for full publication in The Cryosphere. There are a number of major points on the approach, method and presentation that need addressing. Indeed since some parts of the description were not clear, it was difficult to fully evaluate the method. Major points are listed below along with suggestions on how to address them. Given the number of major general comments, at this stage I have not included minor technical corrections to wording or Figure presentation. Overall, important aspects that are required include (among other things) utilizing the CMIP6 HighResMIP experiments to assess resolution-related aspects, incorporating multiple ensemble members to assess the role of internal variability and a more in-depth explanation, motivation and development (i.e. relative to other literature) of the scoring method. Indeed one possibility would be to re-formulate the manuscript with a focus on comparing scores across different resolutions in the CMIP6 HighResMIP experiments and less of a focus on projections.

General comments: 1. Overall model evaluation approach The overall approach to model evaluation presented here differs to that of most other similar studies that I am aware of. For example Agosta et al. and Barthel et al. evaluate CMIP5 models based on selecting those most appropriate for driving regional models and not explicitly on the CMIP5-simulated SMB / surface climate. The main reason for this is that the low-resolution GCMs are not able to realistically capture the correct processes and features associated with the steep orography near the Antarctic coast. This is the job of the regional model and the global model provides the bigger-picture climate responses. In this discussion paper the authors take the approach of explicitly evaluating global standard-resolution models (CMIP5 and CMIP6) directly on their representation of SMB instead of the broader approach taken in the above mentioned studies. They then present projections of SMB change directly from the global models. This presents a number of issues, which are listed below. In addition there are questions on the description and choices made in the methods used, which are also outlined below. Overall, key recommendations are to utilize the CMIP6 HighResMIP experiments to assess resolution-related aspects, to incorporate multiple ensemble members to assess the role of internal variability and explain and motivate the scoring method (and relate to a wider range of existing literature).

1.1 Comparing GCMs with reconstructions A major issue with comparing standard resolution GCMs and observations/reconstructions, is that full GCMs are not able to reproduce the detail required in regions of high precipitation. Therefore a standard-resolution

GCM that reproduces observed/reconstructed Antarctic-wide time-mean SMB is quite possibly doing so for the wrong reasons. This therefore may not be the most appropriate model for projections. The authors should utilize the HighResMIP dataset to determine the resolution dependence of participating models and the potential implications this might have on model selection. This is relevant to all 5 of the criteria used (mean SMB, SMB variability, SMB trends, modes of variability (EOF analysis) and variance explained by the modes). With regard to the EOF analysis, from Figure 5 seems to suggest highly regionalized nature of patterns from the reconstructions. Indeed, an assessment of natural variability is again crucial here in identifying uncertainty in comparing observations and models.

1.2 A lack of mechanistic explanation for why each of the 5 criteria are relevant for improving reliability of projections Firstly the authors should outline the rationale for inclusion of each of the criteria and how they may potentially improve reliability of projections. It is important to discuss this in the context of existing literatures. For example, Krinner et al. (2014) found that future change in SMB was more associated with thermodynamic, rather than dynamic, factors. Secondly the authors should consider the possibility of leave-one-out cross validation, whereby the real world is can be replaced by each member of the model ensemble in turn to see whether evaluation against that model can help improve predictions from that model. This can help to identify which criteria are most relevant in terms of future projections.

1.3 The methodological framework for model weighting In addition to the criteria selected, the rationale for the methodology on model weighting needs to be carefully introduced and motivated. Indeed it is common for a model weighting method to be developed initially in a separate paper and then applied to model output in subsequent papers. Specific suggestions are: Firstly the authors need to bring in more of the previous substantial literature on model weighting. Agosta et al. (2015) use a Climate Prediction Index approach which, as I understand it, draws from probability theory and the probability that observations and models may agree (this goes back to Murphy et

al., 2014). There are also detection and attribution approaches, which use past trends to scale future projections and should be mentioned. What is the advantage of the approach used in this discussion paper? Secondly, the authors should consider the implications of situations where the reconstruction uncertainty is small. In the extreme case where it approaches zero, in general models would be many multiples of this uncertainty range away from the reconstruction. How is/would this be handled in terms of relative weighting across different criteria? Thirdly, the method needs to be described more clearly and is in fact difficult to fully evaluate. The whole section needs to be improved and I have just identified one example starting on line 109. Specifically the text: "if a model time series was fully captured within $2\times$ the reconstruction uncertainty, the model would receive a score of 2". I could not find a clear definition of "reconstruction uncertainty". This exact term is only referred to once in the preceding text on line 69. Is it the same as the "total uncertainty" mentioned on lines 72/73? If so, how does the spatial and temporal information map of total uncertainty map onto the AIS-integrated SMB? In the same paragraph it is not clear what is meant by "model time series fully captured"? Does this mean that even extreme years in the model time series are considered? My recommendation is to write out these score criteria as equations to make it easier for the reader to understand and assess them.

1.4 The role of internal climate variability in trend and spatial EOF analysis The potential role of internal climate variability in evaluating trends is not mentioned, but could be very important. This could be very important for 50-year trends and the spatial EOF patterns. The authors should test the possible role of internal variability by assessing climate models with multiple ensemble members of their historical runs.

1.5 Final model selection The final selection of 4 CMIP5 models for projections should be compared and contrasted with related studies, Agosta et al., (2015) and Barthel et al. (2020). The reasons for, and implications of, differences should be discussed. What is the significance of the smaller spread across these four models. They come from only two model centers (GISS and MPI). Such close links calls into question the

statistical significance of spread across models from just two groups. This could be small or large by chance.

1.6 Impacts of wider factors on projections (e.g. conditions over the Southern Ocean). Another major caveat with the SMB-focused model evaluation is that wider model biases that are known to be important for projections, such sea-surface conditions surrounding Antarctica and hemispheric-scale atmospheric circulate biases, could have an effect on projections (e.g. Krinner et al., 2014; Kittel et al., 2018). The authors do acknowledge this, but don't make implications of differences clear. Could it be that the results of this study should be interpreted alongside other studies?

1.7 Inter-annual variability in GCMs and regional models. On line 79 it is stated that "Global climate models tend to show higher skill at representing interannual variability compared to regional climate models (Medley and Thomas, 2019)". It is not clear to me why this should be since regional models derive their variability from global models. It is also then notable that all CMIP5/6 models over-estimate SMB variability by so much (line 197). An explanation needs to be provided for this, or at least a discussion of the point.

References Agosta, C., Fettweis, X., & Datta, R. (2015). Evaluation of the CMIP5 models in the aim of regional modelling of the Antarctic surface mass balance. The Cryosphere Discussions, 9(3), 3113–3136. https://doi.org/10.5194/tcd-9-3113-2015

Kittel, C., Amory, C., Agosta, C., Delhasse, A., Doutreloup, S., Huot, P.-V., Wyard, C., Fichefet, T., and Fettweis, X.: Sensitivity of the current Antarctic surface mass balance to sea surface conditions using MAR, The Cryosphere, 12, 3827–3839, https://doi.org/10.5194/tc-12-3827-2018, 2018.

Krinner, G., Largeron, C., Ménégoz, M., Agosta, C., & Brutel-Vuilmet, C. (2014). Oceanic Forcing of Antarctic Climate Change: A Study Using a Stretched-Grid Atmospheric General Circulation Model. Journal of Climate, 27(15), 5786–5800. https://doi.org/10.1175/JCLI-D-13-00367.1

Murphy, J. M., Sexton, D. M. H., Barnett, D. N., Jones, G. S., Webb, M. J., Collins, M., and Stainforth, D. A.: Quantification of modelling uncertainties in a large ensemble of climate change simulations, Nature, 430, 768–772, doi:10.1038/nature02771, 2004.

Genthon, C., Krinner, C., & Castebrunet, H. (2009). Antarctic precipitation and climate-change predictions: Horizontal resolution and margin vs plateau issues. Annals of Glaciology, 50(50), 55–60. https://doi.org/10.3189/172756409787769681.
* * *
**TCD**

Interactive
comment

---

## Referee Comment (RC3) · Anonymous Referee #3 · 29 Jan 2020

Review The Cryosphere Manuscript # tc-2019-240.pdf Title: Scoring Antarctic surface mass balance in climate models to refine future projections Authors: Tessa Gorte, Jan T. M. Lenaerts, and Brooke Medley

Summary This submission presents a new method for the evaluation of Antarctic snow mass balance (SMB) in climate models. The method described is new at least in part as it compares the climate model output with the relatively new Antarctic Ice Sheet (AIS) SMB ice-core based reconstructions of Medley & Thomas (2019) rather than atmospheric reanalysis products. The models compared consist of CMIP5 and the few CMIP6 models available at the time of submission. AIS SMB from the climate models was compared with the reconstructed AIS SMB over 1850-2000 and in five categories: AIS integrated mean, trends, temporal variability, and two categories of

spatial variability estimated via empirical orthogonal function (EOF) analysis (pattern and variance represented by the pattern). The best scoring models were NASA's GISS models and the Max Planck's MPI model.

The work described in this article represents an important scientific contribution to evaluating the ability of climate models to accurately simulate AIS SMB. I found, however, that the paper as written made it challenging for me to understand and retain the primary points. Much of the paper reports more details than necessary (burdening the reader) without adequately summarize the main point. Numerical results are presented as a string of numbers – and I get lost reading them. Many of the quantitative results could be better presented with an accompanying table and in the text a concise written summary. I also found that the section in the results discussing different sensitivities to different emission scenarios is unclear and needs more analysis/explanation (see below). As such I recommend consideration for publication with major revisions.

Here are some examples of unnecessary details/weak summary:

Section 2.1 SMB reconstructions This section summarizes the methods used by Medley & Thomas in creating their ice-core derived SMB reconstructions. I found that I was confused by how these were created, as if all the details might be correct but without the "big picture" context. Once I read the abstract for Medley & Thomas, however, I understood. This section can be re-written (and shortened) to better summarize the reconstructions. If the reader wants all the details of the SMB reconstruction, he/she can refer to Medley & Thomas for that.

Section 4 Results Lines 167-168: "The interquartile ranges for CMIP5 and CMIP6 are 1727 to 2282 Gt yr$-1$and1728 to 2229 Gt yr$-1$, respectively, with means of 1940 Gt yr$-1$and 2115 Gt yr$-1$, respectively." What is the take away? For example, something like "CMIP5 models tend to have a slightly smaller mean AIS SMB with a larger range than the CMIP6 models (Table XXX)". The figure shows this, and a table could present the quantitative results for any readers that want them. Similarly for the other results

throughout this section.

Section 4 results: AIS SMB sensitivities to changes in temperature

Lines 294-297 "Comparing the projected change in SMB per degree warming between the emission scenarios gives median sensitivities of 64±80 GtâŮęC−1, 57±33 GtâŮęC−1, and 78±15 GtâŮęC for RCPs 2.6, 4.5, and 8.5, respectively, for the best scoring models. Combined, these data tell us that for stronger emission scenarios, the AIS SMB response will be stronger in both magnitude and trend." The results do not back up this claim. The mean sensitivity for RCP4.5 is lower than that for the RCP2.6! Furthermore, there is no indication here if the differences in the means are statistically significant or not. If model sensitivities of AIS SMB-Temp change with different scenarios – this is a very interesting result (and needs to be backed up better if it is your result – with some explanation to the apparent contradiction of the RCP4.5 having the lowest sensitivity – or maybe there's a typo?). If so, some discussion about what mechanism might explain this. For example, AIS SMB is driven by precipitation and evaporation/sublimation. Are there processes in changing climate that might drive changes in precipitation in addition to changes in temperature? Changes in synoptic weather patterns? Or? Do sensitivities of AIS-SMB to changes in CO2 remain same in all scenarios or do these change? (or do changes in CO2 combine temperature and precipitation sensitivities into "one" proxy for these?)

Minor revisions and notes Line 71 "calculated spatial sampling uncertainty is based" should be "calculated spatial sampling uncertainty based"

lines 84-87 How many CMIP6 models? Later it is claimed that there were so few CMIP6 models available that statistics are not robust for that set….yet the numbers here (53 models, 28 independent and of these 30/19 are CMIP5 which leaves at least 20 CMIP6?).

Line 114 Repeat 1850-2000…think you mean 1950-2000 in second instance

Language is a bit cumbersome and over the top in 3.1 (AIS-integrated SMB criteria)

Got lost again in 3.2 Maybe a couple equations and a map (example) would help. I have the sense it's pretty straightforward but description overcomplicates

Figure 2. Can't see dots in Figure 2B (they overlap too much?)

Line 190 Just because there are fewer models does not necessarily imply that the spread in trends will be less! For example, one could pick CMIP5 models and only use a subsampling and still get same spread if the models selected have large range in trends. Line 200 Not only melt and discharge distributed unequally, but also accumulation (precipitation)!

Lines 213-216 If using place names, have a map showing where these are

Lines 235-236 already defined RCP earlier, no need to do so again here...

Conclusions The recent and similar work of Barthel et al (2019) is mentioned (lines 45-49). Bartel et al was addressing a related albeit slightly different question (than this submission), namely "which climate models would best be used to force a stand-alone ice sheet model?" and compared climate model output to atmospheric reanalysis products. Did their suggestions (best models for stand alone Antarctic Ice Sheet forcing) differ than yours (best models for AIS SMB in the coupled system) or were they similar? Why do you think that is? (perhaps in conclusions – and only need a couple of sentences). Essentially tie in the results of this submission to other current related results.

Figure 4. condense A-D onto one figure

Figure 5 Reconstruction EOFs are low enough that on scale plotted hard to see patterns. Recommend a different scale for reconstruction (and point out in figure caption). Also to help clarify, only need one legend for reconstruction (if you re-scale) and one for 6 panels of model (do not need 9 identical legend bars - extraneous). This will simplify.

Figure 6 Yellow x's very difficult to see. Make more visible.

Figure 7 Hard to see differences from different scenarios (and until 2006 they are identical). Find a way to combine these three panels into one – this will give same information and also new, comparative information

---

## Author Comment (AC1) · 25 Mar 2020

**Reviewer #1**

We thank the reviewer for their insightful and thorough feedback. We found these comments incredibly thoughtful and helpful to ensuring that this paper is of the quality expected for the Cryosphere and the field at large. To address some of the reviewers most major comments, we are reforming our EOF introduction, discussion, and analysis, adding in Fig. 1 and eq. (1), and changing the way we are assessing the temporal variability criterion by switching to the original reanalysis data set here.

**Major**

[Figure]

**"To score the time series magnitude, we assigned a score, x, for how many x-times the reconstruction uncertainty was required for the entire time series to be within the reconstruction uncertainty."**
*\* I think you should reformulate this sentence in a more mathematical framework. What did you code? What is the minimum value of your score, 0 or 1?*
*\* if I understand well, you did max(abs(Model - obs))/(reconstruction uncertainty)? So you scaled the maximum difference of model to obs with the reconstruction uncertainty? Why not using the RMSE scaled by the reconstruction uncertainty?*

Addressing the first point, we agree that the wording here is tricky and we find it difficult to express this code in words alone. As such, we will add Fig. 1 (see below) to the supplementary material (with additional text) to help illustrate our point. We will add that "the minimum possible score, then, is one, for a model who represents SMB that fits entirely within $1\times$ the reconstruction uncertainty."

**"involves finding what spatial SMB patterns explain the highest variance in the AIS integrated SMB time."**
*\* are you sure this what EOF do? Is is not the variance of space-time SMB variability?*
*\* time series (typo?)*

To the first point, EOFs map the spatial pattern of a variable associated with the highest temporal variance of another variable; in this case, we map the spatial pattern of sea level pressure associated to the highest variability in SMB integrated over the AIS. To the second point, the typo will be corrected.

**"To avoid manually sorting the top three modes of variability for all 53 models, we generated difference maps between each of the top three reconstructed modes and each of the top three modes for each model:"**
*\* why do you do this only for the top 3 modes of each model and not e.g. the top 10?*
We will add text to the effect of "the top three modes explain roughly 76% of the total variance explained. The fourth mode explains only about 6% of the total variance and all other modes explain <5% of the total variance. As such, we only include the top three modes in our analysis."

**"We then sorted the top modes of variability for each model based on smallest difference"**
*\* what do you call "the smallest difference"? Do you average absolute differences over the map? Do you compute a RMSE?*

We will include the following text to clarify this point: "For each grid point, we took the absolute value of the difference between the model and the reconstruction. We then summed those differences to generate a single number ("difference number") that represented the difference between the model and the reconstruction in terms of spatial variability. Mathematically, this looks like:

$$\text{difference number} = \sum_{lat} \sum_{lon} |\text{reconstruction}_{lat,lon} - \text{model}_{lat,lon}| \tag{1}$$

We did this for all nine combinations of model and reconstruction maps for the top three modes variability (model$_1$:reconstruction$_1$, model$_1$:reconstruction$_2$, model$_1$:reconstruction$_3$, model$_2$:reconstruction$_1$, model$_2$:reconstruction$_2$, etc.). For reconstruction mode 1 (reconstruction$_1$), then, we matched which model mode best represented this spatial variability by sorting the model modes based on the smallest difference number. We did this for each reconstruction mode (excluding previously matched model modes) to sort the modes based on the smallest difference.

**"After compiling scores for all five of the aforementioned scoring criteria, we normalized each set of scores to be on a scale from one to ten to ensure that**

**each criterion was equally weighted."**
*\* So, if I understand well, you divide each criteria by the max of the criteria? This scaling is extremely sensitive to outliars. You should consider scaling by the interquantile range or by the standard deviation of each of you criteria.*

This is a very good point. We will do a similar scaling by the interquartile range and compare the results. Should the results differ greatly to those done in the current fashion, we will change the paper results to reflect this new method and move the results obtained using the original method into the supplementary material and note how different methodology affects the final outcome.

**"To refine the scope of what we predict for SMB in the future, we used a subset of models that had a final score in the top 10th percentile of CMIP5 and compared them to the entire scope of CMIP5"**
*\* I am not sure it is a correct method. How much is your method sensitive to the number of models you keep? Why do you use this "10th percentile" criteria? I think that 4 models is too little to compute a robust statistic. Is it statistically correct to compare 4 members to 30 members? You should consider e.g. ensemble regression based on models' scores (Bracegirdle and Stephenson, 2012, doi: 10.1007/s00382-012-1330-3)*

Thanks for this suggestion. We did some analysis on how sensitive this result is to the choice of what we define as our 'top models'. To include this point in the text, we will add words to the effect of: We ran a Monte Carlo simulation in which four random CMIP5 models were selected 100,000 times. Those 100,000 sets of four random scores were compared to the four best scoring model scores using a two-sided t-test. From this, we found that, to a 95% confidence level, we can reject the null hypothesis that the four best scoring models are not statistically significantly different from any random four CMIP5 models.

**Figure 1.**
*\* when I see the spatial pattern of trends in 1B and 1C, I wonder why you use a criteria for SMB-integrated values instead of comparing spatial maps of trends? I think using spatial maps of trends would be more relevant.*

In our analysis, we made this separation by first analyzing the AIS-integrated trends and variability, and then focus on the spatial pattern of variability, and how the trend is spatially variable, on sub-ice sheet scales using EOF techniques. As Figure 5 shows, one of the dominant modes of variability in the reconstruction is reflective of the trend shown in Figure 1B, and criteria 4 and 5 score the ability of the models to simulate that pattern.

**"Looking at multiple time "slices" allows us to investigate if models capture the reconstructed SMB trends for the whole time series compared to more recent decades. Here, we looked at three time slices: the entire over-lapping time series from 1850-2000, the last century from 1900-2000, and the last 50 years from 1950-2000."**
*\* I understand that simulating correctly the trends for 1950-2000 may be useful because it quantifies if the global climate models are able to simulate correctly the response to anthropogenic forcing. However I don't think that scoring the trends over the century is useful for your purpose. Your uncertainty on century-scale trends is very small and I wonder if it is not underestimated. It seems difficult to estimate century-scale internal variability from a 200 year reconstruction in fact.*

We appreciate this comment here regarding the long-term variability of SMB. There is a difficult balance, we feel, in selecting the correct time scale for doing this trend analysis. As the reviewer points out, the last 50 years is useful for quantifying the anthropogenic forcing, but the interannual variability over this time period makes for a very large trend uncertainty. The century-length timescale loses this forced response
aspect, but the trend uncertainty is greatly reduced as the reviewer points out. The method for assessing our trend uncertainty is outlined in the text: we performed Monte Carlo simulation wherein we assumed a normal distribution where $\sigma$ = reconstruction uncertainty of possible SMB values for each year. Then we created 10,000 potential SMB time series by choosing SMB values based on that normal distribution for each year and recalculated the trend for each of these time series. Our uncertainty, then, was the standard deviation of this range of trends done in the same method as published by Medley & Thomas in 2019. Basically, while the anthropogenic signal is concentrated in the latter half of the century, the longer time slices confirm the robustness of the trends as the period length increases. The variability overwhelms the signal at shorter period lengths, which results in large uncertainty bounds. By looking at several time slices, we ensure consistency between the model and reconstruction over different intervals. It is equally important to confirm that pre-1950, the trends are relatively small.

**"All CMIP5 and CMIP6 models overestimate SMB variability. The CMIP5 and CMIP6 models range from overestimates of 144% to 261% and 151% to 217% of the reconstruction standard deviation, respectively"** *\* A strong warning here. I have doubts on the reliability of the reconstruction for interannual variability. How does the reconstruction interannual variability compare with the reanalyses variability for the common period? I suspect that the annual accumulation signal extracted from ice cores is dampened.*

We thank the reviewer for this comment. We performed further analysis on the reconstruction interannual variability and compared it to the original reanalysis interannual variability at the 53 ice core sites. Through this analysis, we found that the reconstruction does, certainly, underrepresent interannual variability compared to the reanalysis by a factor of about 1.7. The variability in the reconstruction can only be as large as the variability in the ice core records. Thus, variability can be

heightened or dampened depending on ice core sampling. Further analysis of a synthetic reconstruction that uses the reanalysis P-E time series (rather than the ice cores) suggest that indeed our sampling has biased variability low. Thus, we will evaluate the variability in CMIP5/6 models using the reanalysis data rather than the reconstruction. While reanalyses struggle with trends and magnitudes, Medley et al. (2013) showed that they sufficiently reproduce the interannually variability with high skill.

**"This dipole corresponds to variability in precipitation generated by variations in the track and strength of the Amundsen Sea Low. The Amundsen Sea Low, which represents the pole of circulation variability in Antarctica (Turner et al., 2013), is marked by high precipitation around the coast of the Antarctic Peninsula (Grieger et al., 2016)."**
*\* All this sentence is strange. It is more a discussion than a result.*

This is a very reasonable point. We will move the discussion of the underlying causes for the patterns seen in the EOF analysis to the discussion section of the paper.

**"The Amundsen Sea Low, which represents the pole of circulation variability in Antarctica"?**
*\* What is a pole of circulation variability?*

We will change the wording here to reflect a more accurate description of the Amundsen Sea Low to the effect of: The Amundsen Sea Low, a dominant synoptic phenomenon that drives a significant amount of the circulation variability in West Antarctica and on the Antarctic Peninsula...

**"The second mode of variability represents high variability in West Antarc-**

[Figure]

**tica and the Antarctic Peninsula. This could be caused by the topography in these regions which can induce large amounts of snowfall."**
*\* I am not sure that you interpret the EOFs correctly. The spatial pattern of an EOF associated to its time series explains to a certain amount of the space-time variability, but it does not mean that where the EOF spatial pattern is high there is a high variability.*
*\* " This could be caused by the topography in these regions which can induce large amounts of snowfall." I don't understand why?*

We thank the reviewer for catching this misrepresentation of EOF analysis. The above statement is incorrect in that, high values in the EOF map do not indicate higher variability but rather how much variability that region explains. We will remove the false statements and replace them with words to the effect of: high values on the EOF map indicate regions that explain large amounts of the variability in AIS SMB. Previous work by Scott Hosking et al. (2013) and Turner et al. (2012) (among others) have shown that variability in the Amundsen Sea Low is responsible for large amounts of precipitation variability in West Antarctica and on the Antarctic Peninsula. Because this region dominates the overall AIS precipitation signal (as East Antarctica sees little snowfall by comparison), a variable Amundsen Sea Low signal, here, would explain the EOF pattern reflected in mode 2 of the reconstruction.

**"By comparison, one of the better scoring models for the EOF map criterion, CMCC CM, also shows a dipole between the Antarctic Peninsula and the Ross Sea region for the top mode as well as strong variance signal around the Antarctic Peninsula for mode 2 and a quadrupolar pattern for mode 3."**
*\* When looking at Fig. 5, EOF modes from the two climate models do not ressemble the reconstruction EOF modes, even for the best performing model (row B). Maybe showing the patterns with the same sign as for the reconstruction modes will help (multiply by -1 the climate model patterns). But still, they will remain very different.*

*E.g. in row B there is no high spot near Davis for EOF 3, and there is a large dipole in WAIS. Are you sure of your computation? If yes, are you sure your analysis is relevant?*
*\* What are the biases of the best scoring models for the large scale circulation fields (e.g. sea level pressure over Southern Ocean) over the last 40 years?*

To the reviewer's first point, we can multiply the model EOFs by -1 to make the comparison easier, but, generally, we think that the main point here is not that the models match perfectly with the reconstruction EOF, but rather that it's more about the general regional patterns than local phenomena. No model will perfectly recreate the the regional specifics of the EOFs, nonetheless those on a more local scale, due to the fact that no model is fully capable of perfectly recreating real world physical parameters. To the reviewer's second point, while we find this question interesting, we feel it is beyond the scope of this work which focuses on determining SMB performance based on a select set of scoring criteria related to the Antarctic Ice Sheet proper.

**Fig 9 and associated text :** *\* The climate sensitivity for SMB must be shown in % K-1, because SMB varies exponentially with temperature. You should revise the end of section 4 with regard to climate sensitivities computed in % K-1.*
*\* Given the issues on the scoring and the relevance of selecting four models, the new version of the manuscript might give different results.*

To the first point, we will change everythig to $K^{-1}$ for consistency. To the second point, yes, given the sensitivity of the models to the scoring criteria, changes to these criteria could easily result in a different conclusion as to the top four scoring models. We will make sure that any changes to the scoring regimes that result in changes in the top four scoring models are duly noted in the text.

**Minor**

**"Integrated over the grounded Antarctic ice sheet (AIS), the blowing snow and runoff terms are negligibly small (Lenaerts et al., 2012a)."**
*\* Drifting snow sublimation is still not well modeled and evaluated. You should reformulate, e.g. something like "we neglect blowing snow and runoff and estimate SMB as precipitation minus sublimation"*

We will change this sentence to "We neglect blowing snow and runoff and estimate SMB as precipitation minus sublimation."

**"Over longer time scales"**
*\* Which ones?*

We have added "Over longer ($\sim$100-1000 year) time scales..."

**"The strong regional variability suggests an important impact of variations in synoptic scale patterns around the AIS (Fyke et al. (2017); Marshall et al. (2017))."**
*\* It is known that synoptic scale patters drive the accumulation variability, reformulate, e.g. "Synoptic-scale variability induces a strong regional variability of the SMB"*

We will change the sentence to "Synoptic-scale variability induces a strong regional variability of the SMB."

**"Additionally, as the atmosphere has been warming over large parts of the AIS and can hold more moisture per the Clausius-Clapeyron relation, SMB is expected to show an overall increase"**
*\* Previdi and Polvani (2016, https://iopscience.iop.org/article/10.1088/1748-9326/11/9/094001) state that "the forced SMB increase due to global warming in recent decades is unlikely to be detectable as a result of large natural SMB variability".*

*Your sentence is unclear and potentially wrong for the last decades. Modify and add references.*

We thank the reviewer for catching this clunky language here. The point we were trying to make relates to future SMB rather than that of the past. We will rewrite this sentence to reflect this more accurately to the effect of: Additionally, as the atmosphere is projected to warm both globally and especially in the polar regions, the atmosphere is expected to be able to hold more moisture per the Clausius-Clapeyron relation. As such, SMB is expected to show an overall increase. In recent decades, this forced SMB response is undetectable due to the significant natural SMB variability (Previdi & Polvani (2016)). Teasing apart the forced response from natural SMB variability requires longer SMB time series – on the order of centuries. In 2017, Thomas et al. found no significant SMB trend over the last 1000 years. In 2019, however, Medley & Thomas found that, over the past 200 years, there is a statistically significant SMB increase that can be derived from ice core measurements.

**"but many of those models tend to overestimate annual precipitation values due to poor representation of coastal topography"**
*\* Are you sure it is because of the poor representation of coastal topography?*

We will add that it is likely due to poor representation of coastal topography as previous studies have shown this to be a significant factor in how precipitation is represented of the AIS. We will also add the reference: Genthon et al. (2009) doi:https://doi.org/10.3189/172756409787769681

**"This allows the atmospheric moisture to penetrate too far inland and leads to excessive precipitation on much of the grounded AIS, while underestimating precipitation nearby the coasts (Lenaerts et al. (2012b))."**
*\* I did not read again this article, but it is about "Modeling drifting snow in Antarctica*

*with a regional climate model: 1. Methods and model evaluation", so I am not sure it
is the right paper to cite here? Do you have other references to show that resolution is
the most important factor for modelling Antarctic precipitation?*

We will add references including Palerme et al. (2019) and (2017) here that
better reflect recent studies of Antarctic precipitation patterns in climate models –
including, specifically, CMIP5.

**"Barthel et al. (2019) investigated the Ice Sheet Model Intercomparison
Project version 6 to determine a recommendation of which models to use for ice
sheet model forcings based on best captured current Antarctic climate relative
to observations and their ability to project certain metrics into the future"**
*\* It's "Ice Sheet Model Intercomparison Project \*for CMIP6\*" and not "version 6" (in
fact it's version 1).*
*\* Barthel et al. (2019) evaluate the global climat models based on their ability to
capture the large scale circulation around ice sheets compared to reanalyses. It is
not "very similar" to your study because the "observation" they use is well evaluated
(reanalyses large scale fields after 1979) and they don't use this criteria to constrain
future projections.*

Addressing the first point: we will fix this typo as follows: Barthel et al. (2019)
investigated the Ice Sheet Model Intercomparison Project for CMIP6 to determine...
Addressing the second point: we will rephrase this sentence to the effect of: The
object of this paper is similar in that Barthel et al. (2019) use scoring criteria to refine
model selection specifically for ice sheet model forcing. Their work differs in that
their criteria look more at the large-scale circulation patterns around ice sheets and
the data set to which they compare models consists of large-scale fields reanalysis
fields. Additionally, they don't then use this subselection of models to constrain future
projections.

**"To improve upon model estimates, several groups have combined ice core data with models to create spatio-temporally robust SMB data sets (Monaghan et al. (2006), Thomas et al. (2017), Medley and Thomas (2019))."** *this sentence should be in the Method section*

We will move this sentence to the methods section.

**"In this work, we leverage the availability of that new avenue for climate model evaluation of AIS SMB, and compare the suite of CMIP5 and CMIP6 climate models to that new SMB reconstruction."** *repetition of the sentence P2 L50-52, merge the two.*

We will merge these two sentences to avoid repetition.

**"they weighted each ice core spatially to generate the 200-year data set"**
*give the period*

We will change this sentence to: they weighted each ice core spatially to generate the 200-year (1800-2000) data set

**"they calculated spatial sampling uncertainty is based on the RMSE"**
*"they calculated spatial sampling uncertainty based on the RMSE"*

We will correct this typo.

**"Global climate models tend to show higher skill at representing interannual variability compared to regional climate models (Medley and Thomas, 2019)."**

*\* it is not what is said in Medley and Thomas, 2019. They say "Because of their aforementioned ability to reproduce the interannual variability[17], which strengthens the weighting scheme, we used \*global atmospheric reanalyses\* over regional climate models.". So this statement is for \*reanalyses\* compared to RCM only, and is based on [17] Medley, B. et al. Airborne-radar and ice-core observations of annual snow accumulation over Thwaites Glacier, West Antarctica confirm the spatiotemporal variability of global and regional atmospheric models. Geophys. Res. Lett. 40, 3649–3654 (2013).*

We thank the reviewer for catching this error here. This is absolutely correct and we will remove this sentence and adjust the following sentence to stress our other main reasons for using global climate models for comparison including that we want to compare future model output to the end of the 21st century for which GCMs are necessary and that this work is meant to guide the selection of GCMs for ice sheet modelers to investigate the global impacts of changing ice sheets.

**"To get a comprehensive look at how well global climate models capture SMB, we compared the suite of CMIP5 models to the reconstruction."**
*\* and CMIP6?*

We will change this sentence to: To get a comprehensive look at how well global climate models capture SMB, we compared the suites of CMIP5 and CMIP6 models to the reconstruction.

**P4 L90-95**
*\* I am not sure the detail of conversion of kg m-2 s-1 in Gt yr-1 is useful. Just saying that it is computed on the original GCM grid is enough.*

We will remove the details regarding unit conversion for succinctness.

**P4 L99-100**
*remove parentheses*

We will remove the parentheses.

**P4 L107: "the magnitude of the SMB time series"**
*\* do you mean "the SMB mean value"? If yes it seems clearer for me to replace "magnitude" by "mean value" everywhere.*

We will change magnitude to mean value throughout the document and add a sentence explaining explicitly what is meant by "mean value."

**"To achieve this goal, we analyzed trends from 1850-2000, 1900-2000, and 1850-2000."** *\* typo*
*\* how do you combine the 3 periods?*

There is a typo here: the last time period should be 1950-2000. Beyond that, we are not sure that we understand the reviewer's question.

**"To score the time series variability, we detrended and normalized each time series to separate the SMB trend from its absolute magnitude using:"**
*\* I don't understand "to separate the SMB trend from its absolute magnitude"*

This is a typo and will be changed to: to separate the SMB variability from its absolute magnitude... We will also add a couple of sentences clarifying what this means to the effect of: if a model should greatly underestimate the mean value, for example, the variability about that mean value will also likely be underestimated. To ensure that we are not double-counting the impact of SMB mean value, we calculate

the variability about the normalized time series.

**"To do so, we performed an empirical orthogonal function (EOF) analysis"**
*\* on annual data over 1850-2005(?)*

We will add this information to this sentence to the effect of: to do so, we performed an empirical orthogonal function (EOF) analysis on annual data over 1850-2000.

**"By breaking this criterion down into two main factors, we were able to determine the models' abilities to accurately capture the modes of variability as well as how much variance each mode explained."**
*\* what are the two main factors you are talking about?*

We will expand this sentence to include: By breaking this criterion down into two main factors, spatial variability and variance explained, ...

**P5 L169 "All four of best scoring models are captured within the reconstructed uncertainty for the entire 150 year time series."**
*\* After reading further I understood that the best scoring models are for the combination of criteria. I think you should begin your result section by presenting the best scoring models (currently presented P10 and in the Figures' legends)*

We will add a paragraph at the beginning of the results section explaining the selection of the top four scoring models and listing the models. We will try to make abundantly clear in the text that the result of the top scoring models that appear throughout Figures 2-5 were added in retroactively to show how well these models do for each criterion in comparison to the rest of the ensembles.

**"We weighted all scores from the five scoring criteria equally on a scale from 1 to 10 with lower scores indicating better performance. The final score, then, is the sum of all the individual scores, which is renormalized on a scale of 1 to 10 with lower scores still indicating better performance."**
*\* repetition of P5 L141-143*

We will remove this repetition.

**" The reconstructed AIS SMB averaged from 1801-2000 shows higher SMB values around the coastal areas, particularly in the Antarctic Peninsula and West Antarctic regions (Fig. 1A)."**
*\* This is really the most basic feature of Antarctic SMB, this sentence is not useful*

We will remove this sentence to keep the section concise.

[Figure]

Figures/UncertaintyIllustration.jpg

**Fig. 1.** Time series of the reconstructed AIS-integrated SMB time series (purple) with $1\times$, $21\times$, and $31\times$ the uncertainty in dark purple, medium purple, and light purple, respectively. Three model AIS-integrated SMB time series, MPI ESM LR (green), IPSL CM5A LR (yellow), and BNU ESM (teal) have been plotted as well to demonstrate different model scoring. MPI ESM LR is entirely captured within $1\times$ the reconstruction uncertainty and, thus, receives a score of 1. IPSL CM5A LR is entire captured within $2\times$ the uncertainty so its score for this criterion is 2. BNU ESM is fully captured within $7\times$ the uncertainty.

[Figure]

**Fig. 2.**

---

## Author Comment (AC2) · 25 Mar 2020

**Reviewer #2**

We greatly appreciate the reviewer's suggestions to add to the robustness of our study through the comparison across ensemble members and resolutions within a single model very insightful. These are both very good suggestions that will elevate the scientific quality of this paper. Additionally, we also thank the reviewer for the leave-one-out analysis suggestion. This is not an approach we had considered taking. We also appreciate the references the reviewer listed to help us add context to our study.

[Figure]

**Overall, important aspects that are required include (among other things) utilizing the CMIP6 HighResMIP experiments to assess resolution-related aspects, incorporating multiple ensemble members to assess the role of internal variability and a more in-depth explanation, motivation and development (i.e. relative to other literature) of the scoring method. Indeed one possibility would be to re-formulate the manuscript with a focus on comparing scores across different resolutions in the CMIP6 HighResMIP experiments and less of a focus on projections.**

These are good points, it is important to make sure we address spread within a model via ensemble members and resolution. To address the former, we will use the Community Earth System Model Large Ensemble (CESM-LENS) and score each of its 35 ensemble members. In doing so, we will be able to say whether a single ensemble member is representative of the whole model. If not, we will redo the analysis using ensemble means for the models. Additionally, we will do a similar analysis using low, middle, and high resolution simulations for the HadGEM-GC31 and ECMWF-IFS models. Here, though, our analysis will lend insight into whether resolution significantly impacts our results.

*Comparing GCMs with reconstructions: A major issue with comparing standard resolution GCMs and observations/reconstructions, is that full GCMs are not able to reproduce the detail required in regions of high precipitation. Therefore a standard-resolution GCM that reproduces observed/reconstructed Antarctic-wide time-mean SMB is quite possibly doing so for the wrong reasons. This therefore may not be the most appropriate model for projections. The authors should utilize the HighResMIP dataset to determine the resolution dependence of participating models and the potential implications this might have on model selection. This is relevant to all 5 of the criteria used (mean SMB, SMB variability, SMB trends, modes of variability (EOF analysis) and variance explained by the modes). With regard to the EOF analysis,*

*from Figure 5 seems to suggest highly regionalized nature of patterns from the reconstructions. Indeed, an assessment of natural variability is again crucial here in identifying uncertainty in comparing observations and models.*

There is a lot of validity in saying that GCMs are, compared to RCMs, incredibly low resolution which, thus, makes it difficult for them to reproduce the detailed SMB response. We also feel that RCMs are much more accurate in capturing AIS SMB. However, due to their high resolution, RCMs are also relatively computationally expensive to run for long periods (∼100s of years). Because one of the goals of this paper is to investigate the future of SMB over Antarctica, we use GCMs for their ability to simulate these long-term climate effects. Additionally, as RCMs are by definition regional, they need boundary forcings adding an additional layer of complexity to running the models for century-length timescales.

An additional reason that we decided to use GCMs is simply to figure out which GCMs perform best at capturing this phenomenon. There has been extensive work investigating SMB in RCMs as the reviewer points out, but relatively little looking at GCMs. To investigate the global coupled response to future SMB changes, one needs GCMs, not RCMs. As such, this work is meant to inform modelers who are concerned with global ramifications of changing AIS SMB.

We do not want to dismiss this point, though, that RCMs will almost always be more accurate at representing SMB and its drivers. We will be sure to add text to the introduction section that underscores our reasoning for investigating GCMs over RCMs. Additionally, we will include 2 models each with 3 resolutions from the HighResMIP experiment to help address the valid concerns about the impact of resolution on model score. We will also include in supplementary materials a scatter plot of model resolution versus overall score to further address this issue.

*A lack of mechanistic explanation for why each of the 5 criteria are relevant for improving reliability of projections: Firstly the authors should outline the rationale for*

[Figure]

*inclusion of each of the criteria and how they may potentially improve reliability of projections. It is important to discuss this in the context of existing literatures. For example, Krinner et al. (2014) found that future change in SMB was more associated with thermodynamic, rather than dynamic, factors. Secondly the authors should consider the possibility of leave-one-out cross validation, whereby the real world is can be replaced by each member of the model ensemble in turn to see whether evaluation against that model can help improve predictions from that model. This can help to identify which criteria are most relevant in terms of future projections.*

We understand the reviewers comments here regarding the criteria selection process. This process went through several iterations of internal review and revision. As the reviewer rightly points out, there is preexisting literature investigating underlying thermodynamic processes that drive AIS SMB. However, we feel that our paper is different from these earlier papers in that we are not trying to investigate a models ability to capture these drivers as much as reproduce the actual reconstructed SMB record. We do, with our EOF analysis, check whether models are recreating spatial SMB patterns of variability which, we feel, addresses the point as to whether the models are, on whole, doing a sufficient job of recreating the physical SMB drivers. We do, however, feel that the reviewer also makes a valid point that we could further justify the choices we made in selecting the criteria and cite more preexisting literature. We will also include text that denotes the separation of this work from the other related literature. We also appreciate the suggestion of the leave-one-out cross validation and will perform this analysis with our current criteria in the supplementary material.

*The methodological framework for model weighting: In addition to the criteria selected, the rationale for the methodology on model weighting needs to be carefully introduced and motivated. Indeed it is common for a model weighting method to be developed initially in a separate paper and then applied to model output in subsequent papers. Specific suggestions are: Firstly the authors need to bring in more of the*

*previous substantial literature on model weighting. Agosta et al. (2015) use a Climate Prediction Index approach which, as I understand it, draws from probability theory and the probability that observations and models may agree (this goes back to Murphy et al., 2014). There are also detection and attribution approaches, which use past trends to scale future projections and should be mentioned. What is the advantage of the approach used in this discussion paper? Secondly, the authors should consider the implications of situations where the reconstruction uncertainty is small. In the extreme case where it approaches zero, in general models would be many multiples of this uncertainty range away from the reconstruction. How is/would this be handled in terms of relative weighting across different criteria? Thirdly, the method needs to be described more clearly and is in fact difficult to fully evaluate. The whole section needs to be improved and I have just identified one example starting on line 109. Specifically the text: "if a model time series was fully captured within 2× the reconstruction uncertainty, the model would receive a score of 2". I could not find a clear definition of "reconstruction uncertainty". This exact term is only referred to once in the preceding text on line 69. Is it the same as the "total uncertainty" mentioned on lines 72/73? If so, how does the spatial and temporal information map of total uncertainty map onto the AIS-integrated SMB? In the same paragraph it is not clear what is meant by "model time series fully captured"? Does this mean that even extreme years in the model time series are considered? My recommendation is to write out these score criteria as equations to make it easier for the reader to understand and assess them.*

Addressing the point that justification for the criteria needs to be introduced and/or expanded upon: we will make sure to further rationalize the inclusion of each criteria and add context from the literature.

In regards to the point wherein the reconstruction uncertainty approaches zero, if this is the case, then all the models would score highly on this criterion, that is correct. However, after doing the initial scoring, each criterion score spread is normalized to a scale ranging from 1 to 10. As such, all scoring criteria are weighted equally. We

realize that this needs to be expanded upon in the text, and so we shall be sure to include a more detailed description of this process to alleviate further confusion.

We agree with the reviewers recommendation for added clarity in the text regarding the scoring process. We will add Fig. **??** as well as several equations for relevant criteria to help illustrate the process which, ideally, will offer much more insight into the process.

*The role of internal climate variability in trend and spatial EOF analysis: The potential role of internal climate variability in evaluating trends is not mentioned, but could be very important. This could be very important for 50-year trends and the spatial EOF patterns. The authors should test the possible role of internal variability by assessing climate models with multiple ensemble members of their historical runs.*

To address a very valid comment by the reviewer earlier, we will be including the CESM-LENS experiment to take into account changes due to internal variability. In our analysis of this sub-experiment, we will make sure to address in greater detail the potential role of internal variability both with regard to shorter trends and EOF analysis.

*Final model selection: The final selection of 4 CMIP5 models for projections should be compared and contrasted with related studies, Agosta et al., (2015) and Barthel et al. (2020). The reasons for, and implications of, differences should be discussed. What is the significance of the smaller spread across these four models. They come from only two model centers (GISS and MPI). Such close links calls into question the statistical significance of spread across models from just two groups. This could be small or large by chance.*

We will add in discussion of the results found in both papers, stressing the differences between the methodology and and the influence of these differences on the final results. Both papers reached different final conclusions regarding which

models they concluded captured AIS SMB best due, in large part, to disparities in the methodology. Agosta et al. (2015) focused their investigation into more atmospheric and oceanic dynamics (sea ice extent, sea surface temperature, sea surface pressure, precipitable water, 850 hPa temperature) while Barthel et al. (2020) ruled out both the GISS and MPI modeling groups due to their initial selection criteria. However, we will still explore how the common models compare overall from our analysis compared to their papers. We will also add in further analysis looking at the spreads for each modeling group and note whether the reduction in spread is more a result of modeling group (and, thus, model physics) or a more reflective spread due to uncertainty reduction with regard to AIS SMB.

*Impacts of wider factors on projections (e.g. conditions over the Southern Ocean): Another major caveat with the SMB-focused model evaluation is that wider model biases that are known to be important for projections, such sea-surface conditions surrounding Antarctica and hemispheric-scale atmospheric circulate biases, could have an effect on projections (e.g. Krinner et al., 2014; Kittel et al., 2018). The authors do acknowledge this, but don't make implications of differences clear. Could it be that the results of this study should be interpreted alongside other studies?*

We appreciate the reviewer's comment here that we should include more of this literature to provide context to our experiment. Acknowledging these biases is important for providing a complete picture to the reader as well as putting into context the limitations of this work. We will include in our text words to the effect of: One of the major caveats with this work is the relatively narrow scope of just looking at AIS SMB. Because we refined our criteria at the outset of our experiment to solely reflect model performance with regard to capturing SMB and didn't include outside factors like synoptic weather patterns, sea ice or sea surface conditions (Krinner et al. (201; Kittel et al. (2018)), there are potentially some wider model biases that we are missing that could affect SMB projections. In our analysis, we make the significant assumption

that the past ability to capture SMB correlates to higher skill in projecting AIS SMB into the future. However, model biases in some of the larger physical drivers – and how those biases change into the future – will significantly impact future AIS SMB trajectory.

*Inter-annual variability in GCMs and regional models: On line 79 it is stated that "Global climate models tend to show higher skill at representing interannual variability compared to regional climate models (Medley and Thomas, 2019)". It is not clear to me why this should be since regional models derive their variability from global models. It is also then notable that all CMIP5/6 models over-estimate SMB variability by so much (line 197). An explanation needs to be provided for this, or at least a discussion of the point.*

This is a misinterpretation on our part here. Medley & Thomas point out that global atmospheric reanalyses show higher skill than regional models in capturing interannual variability. We will remove this sentence and adjust the following sentence to stress our other main reasons for using global climate models for comparison. Additionally, we have done further analysis on the reconstruction interannual variability and found that the reconstruction process dampens the actual SMB interannual variability signal by a factor of about 1.7 compared to the original reanalysis P-E data. Our analysis shows that this is predominantly owing to under-sampling of highly variable ice cores in selection process. As a result, for this criterion, we will use the original reanalysis data and make a strong note about the interannual variability issues in the reconstruction. We will still, however, include the reconstruction temporal variability analysis in the supplementary material and note if and how changing the "observational" record for this single criterion impacts the final scoring result. With regards to global reanalysis models, though, Medley et al. (2013) did find that global reanalyses exhibited higher skill at reproducing interannual variability than the Regional Climate Model RACMO2. Further analysis revealed that the lack of upper atmosphere constraint allowed the weather to deviate too far from the driving reanaly-

sis. Van de Berg & Medley (2016) determined that applying upper air relaxation within the RCM provided the necessary constraint and significantly improved the relationship between RCM and observations. Thus, RCMs that use upper air relaxation typically exhibit higher skill in reproducing the interannual variability, so often there is a range of skill depending on how much freedom the RCM is given to deviate from reanalysis forcing.
* * *
[Figure]

[Figure]

**Fig. 1.**

---

## Author Comment (AC3) · 25 Mar 2020

**Reviewer #3**

We thanks the reviewer for providing a lot of thoughtful insight and asking a lot of very good questions. To address the predominant remarks, here, we will make the necessary adjustments to the wording to make the paper more accurate and comprehensible. We have also added Fig. **??** and eq. (1) to help with this process. Additionally, we would like to thank the reviewer for their helpful comments on figure adjustments. We appreciate how important for overall comprehension good figures are and we will strive to make ours as palatable as possible.

[Figure]

**Section 2.1 SMB reconstructions:**
*This section summarizes the methods used by Medley & Thomas in creating their ice-core derived SMB reconstructions. I found that I was confused by how these were created, as if all the details might be correct but without the "big picture" context. Once I read the abstract for Medley & Thomas, however, I understood. This section can be re-written (and shortened) to better summarize the reconstructions. If the reader wants all the details of the SMB reconstruction, he/she can refer to Medley & Thomas for that.*

We will trim down the details of this section and add in more "big picture information" to the effect of: the reconstructions, generated by Medley & Thomas, provide a 200-year record of AIS SMB. The authors synthesize SMB time series from an extensive ice-core database with reanalysis-derived spatial coherence patterns to generate a continent-wide AIS SMB data set. We will move the details about the actual hybridization process of the ice-cores with reanalysis spatial fields to supplementary material.

**Section 4 Results Lines 167-168: "The interquartile ranges for CMIP5 and CMIP6 are 1727 to 2282 Gt yr$-1$ and 1728 to 2229 Gt yr$-1$, respectively, with means of 1940 Gt yr$-1$ and 2115 Gt yr$-1$, respectively."**
*What is the take away? For example, something like "CMIP5 models tend to have a slightly smaller mean AIS SMB with a larger range than the CMIP6 models (Table XXX)". The figure shows this, and a table could present the quantitative results for any readers that want them. Similarly for the other results throughout this section.*

We will add more context for the each result throughout the section including how the different CMIP ensembles compare to one another and where the reconstruction falls in relation to each. We will also add more information, where relevant, about how these values compare to other recent studies within the CMIP5/CMIP6 framework.

**Section 4 results: AIS SMB sensitivities to changes in temperature: Lines 294-297: "Comparing the projected change in SMB per degree warming between the emission scenarios gives median sensitivities of 64±80 Gt C−1, 57±33 Gt C−1, and 78±15 Gt C for RCPs 2.6, 4.5, and 8.5, respectively, for the best scoring models. Combined, these data tell us that for stronger emission scenarios, the AIS SMB response will be stronger in both magnitude and trend."**

*The results do not back up this claim. The mean sensitivity for RCP4.5 is lower than that for the RCP2.6! Furthermore, there is no indication here if the differences in the means are statistically significant or not. If model sensitivities of AIS SMB-Temp change with different scenarios – this is a very interesting result (and needs to be backed up better if it is your result – with some explanation to the apparent contra-diction of the RCP4.5 having the lowest sensitivity – or maybe there's a typo?). If so, some discussion about what mechanism might explain this. For example, AIS SMB is driven by precipitation and evaporation/sublimation. Are there processes in changing climate that might drive changes in precipitation in addition to changes in temperature? Changes in synoptic weather patterns? Or? Do sensitivities of AIS-SMB to changes in $CO_2$ remain same in all scenarios or do these change? (or do changes in $CO_2$ combine temperature and precipitation sensitivities into "one" proxy for these?)*

We would like to apologize here. The reviewer is correct in that the results are not statistically different between the three forcing scenarios. This text is based off an early result that had since been updated in the figure. We failed to update the text as a result and will do so to make sure it accurately reflects this newer result to the effect of: these results indicate that there is no difference in the sensitivity response to changes in temperature between the three forcing scenarios. We will also make sure that this result is changed throughout the document to accurately portray the result presented here.

**Line 71 "calculated spatial sampling uncertainty is based"**

[Figure]

*should be "calculated spatial sampling uncertainty based"*

We will correct this typo.

**lines 84-87**
*How many CMIP6 models? Later it is claimed that there were so few CMIP6 models available that statistics are not robust for that set...yet the numbers here (53 models, 28 independent and of these 30/19 are CMIP5 which leaves at least 20 CMIP6?).*

Because we were performing the analysis as CMIP6 was being released, we only had access to 12 models at the time. We are continuing to add in more CMIP6 models as they are released and we will add them into our analysis and discussion.

**Line 114**
*4 Repeat 1850-2000...think you mean 1950-2000 in second instance*

Will will correct this typo.

*Language is a bit cumbersome and over the top in 3.1 (AIS-integrated SMB criteria)*

We recognize that the language here is difficult to parse and we appreciate the reviewer's comment. We are adding Fig. **??** and will use it when referring to our scoring process. With this, we aim to clarify our scoring method in a concise way.

*Got lost again in 3.2 Maybe a couple equations and a map (example) would help. I have the sense it's pretty straightforward but description overcomplicates*

Again, we agree with the reviewer here that the language lends itself to being
difficult to understand. We are adding eq. (1) to further inform the reader of the process we used to parse the spatial criteria.

**Figure 2**
*Can't see dots in Figure 2B (they overlap too much?)*

Because there are relatively small differences between the top scoring models compared to the entire ensemble, it is difficult to differentiate the dots, yes. However, we do believe that better color choices could be made to accentuate any differences and we will make a note in the caption that some dots may be overlapping.

**Line 190**
*Just because there are fewer models does not necessarily imply that the spread in trends will be less! For example, one could pick CMIP5 models and only use a subsampling and still get same spread if the models selected have large range in trends.*

This is a very good point. We will change our wording to reflect words to the effect of: the spread in trend in the CMIP6 models is significantly lower than for CMIP5 models which is likely due to a smaller sample that isn't capturing the extreme ends of the distribution seen in CMIP5.

**Line 200**
*Not only melt and discharge distributed unequally, but also accumulation (precipitation)!*

We thank the reviewer for catching this. We will change the wording to: ... spatial variations in SMB are also important in AIS SMB representation in models as precipitation, melt, and discharge are not distributed equally.

[Figure]

**Lines 213-216**
*If using place names, have a map showing where these are*

We will add a map to the supplementary material with all place names referenced in the work located with labels.

**Lines 235-236**
*already defined RCP earlier, no need to do so again here...*

We will remove this redundancy.

**Conclusions**
*The recent and similar work of Barthel et al (2019) is mentioned (lines 45-49). Bartel et al was addressing a related albeit slightly different question (than this submission), namely "which climate models would best be used to force a stand-alone ice sheet model?" and compared climate model output to atmospheric reanalysis products. Did their suggestions (best models for stand alone Antarctic Ice Sheet forcing) differ than yours (best models for AIS SMB in the coupled system) or were they similar? Why do you think that is? (perhaps in conclusions – and only need a couple of sentences). Essentially tie in the results of this submission to other current related results.*

We agree that further discussion of this paper is relevant for our conclusions. Barthel et al. were addressing a slightly different question, as the reviewer points out, and their methodology is different as well. One of their first steps was to exclude models whose output was not 6-hourly and models that didn't include both RCP2.6 and RCP8.5 output (as this was a stipulation of the ISMIP6 experiment). With that first step, Barthel et al. already excluded the GISS and MPI model groups which we found to have the best results for CMIP5. As such, making a one-to-one comparison with the

[Figure]

final results of this paper is difficult, but we will address some of the similarities and differences in other model results while stressing the differences in methodology and purpose of the work.

**Figure 4**
*condense A-D onto one figure*

We have tried to combine panels A-D for this figure but have found that the distributions are too similar in width and height that they are largely indistinguishable.

**Figure 5**
*Reconstruction EOFs are low enough that on scale plotted hard to see patterns. Recommend a different scale for reconstruction (and point out in figure caption). Also to help clarify, only need one legend for reconstruction (if you re-scale) and one for 6 panels of model (do not need 9 identical legend bars - extraneous). This will simplify.*

We will update the scale on the reconstruction EOF figure to make the patterns more apparent and make note of the scale change in the figure caption. We will also remove duplicate color bars to simplify the plot and enhance overall readability of the figure.

**Figure 6**
*Yellow x's very difficult to see. Make more visible.*

We will change these x's to make them more visible.

**Figure 7**
*Hard to see differences from different scenarios (and until 2006 they are identical). Find a way to combine these three panels into one – this will give same information*

*and also new, comparative information*

We agree that it is a bit difficult to see accurately the differences between the scenarios in this figure. We have tried multiple ways to convey this information succinctly in a single frame to alleviate this issue but have repeatedly found that our attempts to do so only reduce the readability of the figure. For instance, combining the figure as is into one frame makes it such that the larger model spread (for all models) are difficult – if not impossible – to differentiate due simply to the fact that there is significant spread amongst all the models in every forcing scenario. We can, however, try to add a frame that just looks at the four best models in each of the forcing scenarios to be able to make direct comparison among a smaller subset of models.

[Figure]

**Fig. 1.**

---

## Referee Report (RR1)

Warning: because of my very limited number for working hours these past weeks/months, I only focused on the answers to my comments and to the associated text. I apologize I haven't taken time to re-read the full article, despite my great interest for this work.

First I want to thank the authors for carefully considering all my comments and suggestion, it is much appreciated.

I recommend this article for publication in TC. I have a few minor suggestions, and 2 more important comments, listed bellow :

**\* To the first point, EOFs map the spatial pattern of a variable associated with the highest temporal variance of another variable;**

Are you sure that is true? From https://www.sciencedirect.com/topics/earth-and-planetary-sciences/empirical-orthogonal-function-analysis : « EOFs of a space-time physical process can represent mutually orthogonal space patterns where the data variance is concentrated, with the first pattern being responsible for the largest part of the variance, the second for the largest part of the remaining variance, and so on. »

It seems that EOF maps spatial patterns associated with total (space-time) variance.

**\* in this case, we map the spatial pattern of sea level pressure associated to the highest variability in SMB integrated over the AIS. To the second point, the typo will be corrected.**

Is there a typo? Is it spatial pattern of SMB instead of spatial pattern of sea level pressure ?

\* Because CMIP6 uses a different future forcing scenario mechanism (Shared Socioeconomic Pathways), CMIP5 and CMIP6 future projections are not directly comparable.

**Major:** SSPs were designed to be comparable to CMIP5 RCPs, with indicated RCP in the SSPs: see Fig.2 of https://www.geosci-model-dev.net/9/3461/2016/gmd-9-3461-2016.pdf. Comparison between CMIP5 and CMIP6 is fully included in the next IPCC report.

So you should include CMIP6 in your improved projections.

\* the top 90th percentile overall scoring models were determined to be GISS E2 H CC, GISS E2 R CC, GISS E2 R, MPI ESM LR, MPI ESM MR, and MPI ESM P

**Major:** It's not unexpected that from the 6 « best » models, there are 2 clusters of same modeling center simulations : 3 GISS-E2 and 3 MPI-ESM. I think you must add a criteria in your model selection to select models from different modeling centers, because these models will share the same biaises (see e.g. https://www.pnas.org/content/115/38/9462). I think it is important to sample a diversity in model simulations. Can you remove 2 GISS and 2 MPI and add 2 to 4 other models instead ?

If you don't, you should add a table with all models ranked by the final score and including the score for each criteria and modeled SMB projection for each scenario; ~similarly to what you did in Supplementary but including all criteria scores and projections.

NB: it seems that Tables in Supplementary materials are not correctly displayed.

\* These eight models have been added in retroactively to figures 2-3

I don't see added models in Fig. 2? Maybe it's Fig. 3 and Fig. 4?

\* The reconstructed AIS SMB averaged from 1801-2000 shows Along with higher SMB values around the coastal areas, particularly in the Antarctic Peninsula and West Antarctic regions (Fig. 2A). The highest absolute SMB trends are around the , the coastal regions of East Antarctica and the Antarctic Peninsula also show the highest absolute SMB trends (Fig. 2B). **This** reconstruction also highlights large portions of ...

You have to introduce the reconstruction in the modified sentence.

---

## Author Response (AR2)

[revised manuscript text omitted]

Dear editor,

     During these times, we recognize that it can be extremely difficult to get much work done and that, often, reviewership does not take precedence. As such, we would like to thank you and the reviewers for taking the time to read the revised manuscript and provide further comments.

Thank you for your consideration,
Tessa Gorte and co-authors.

**Reviewer 1**

**Lines 25-27 repetitive, delete one sentence (or the other)**
We have deleted the sentence: "Ignoring these terms, AIS SMB can be estimated as SMB = precipitation - sublimation."

**Line 47 MB = mass balance? GMSL = global mean sea level? Define**
We have changed the sentence to read: "Despite its importance for AIS mass balance (MB) and global mean sea level (GMSL), ..."

**Line 99 MERRA-2 reference?**
We have added "... we will use the MERRA-2 based data set provided by Medley & Thomas (2019) as a proxy..."

**Line 245 "added retroactively figures 2-3 for comparison? Not in Figure 2...**
This typo has been corrected to: "figure 3-5."

**Figure 1. These colors appear blue on my screen (not purple)?**
We have changed all instances of "purple" to "indigo" to alleviate some confusion.

**Figure 2 and text – why are mean, trends shown from 1801-2000 when the model scoring is for 1850-2000? Or am I missing something?**
This figure is meant to show the patterns of means and trends in the reconstruction so we didn't feel that the time span was so important. We take your point, however, that it could lead to confusion, so we have replotted the figure with data only spanning 1850-2000.

**Lines 264-280 Although this is better than the first submission, I still find sections in here become overwhelming and read as a list of numbers. I would prefer having a table presenting the numbers for any readers who want the actual values (this makes it also easier for future work to find values in the paper and cite or compare!) with written text more of a summary (with details in the table). Figure 4 also summarizes quite nicely. This is more of a stylistic recommendation and the authors may choose to disagree and keep as is...**
We do appreciate the complexity in reading lots of lists of numbers. We feel it is important and relevant to have all the numbers listed in the text, but as we take your point, we have also added in Table 1 which provides a list of all the ranges for the temporal criteria as a succinct reference point.

**Line 299 should be "Figure 5A, B" or simply "Figure5"**
We have changed the text to "Figure 5A, B."

**Lines 302-303 states that the STD for the reanalysis is 6.6% and reconstruction 2.9%, but the figure caption states that 6.6% is the reconstruction. Which is it?**

We have changed the typo in the figure caption to: "The black dots show the standard deviation of the original MERRA-2 reanalysis."

**Lines 413-415 It seems like the CESM-LE performs, on the whole, better than the CESM CMIP5 contributions. Why do you think this is the case? A detailed analysis is beyond the scope of this paper – just find this to be very interesting and wonder why this might be (at most 1-2 sentences)? I think there are two possibilities here (although maybe more?). One, there are only 5 individual CMIP5 CESM runs, compared with 35 LE and only one of the 5 is a clear outlier (FASTCHEM). Thus, the differences in scores and ranges may not be significant and only show that scoring itself is susceptible to internal variability (and what a large range!), and cautions against using single-ensembles to estimate model performance. Another possibility may lie in forcing differences. I believe the CESM-LE does not have the same ozone forcing as the CESM CMIP5 contributions, and I would expect SH to be the region this would matter most (I don't recall exactly the difference but I think the CESM LE is much closer to obs for ozone forcing, particularly in the SH?). Also, I am not sure if the shorter overlap of time (LE 1920-2000 and CMIP5 1850-2000) should impact the EOF analysis (the trend, yes maybe). If the modes of variability are robust under strong external forcing, then the LE – with 30 ensembles – should have plenty of samples to adequately estimate EOFS (I'm actually not as convinced of single-ensemble members). The exceptions will be if the EOFs are not robust under anthropogenic forcing, or if there are mulit-decadal modes missing in the CESM LE due to identical ocean initial conditions (although not convinced they would be resolved in single-runs of the different CESM CMIP5 simulations).**

We agree with the reviewer, here, that this is a curious result (and that finding the exact answer to it lies slightly beyond the scope of this work). We do think that this is a strong cautionary tale against using single-ensemble models as they can be very misleading. Generally, though, we think that the forcing (the reviewer's second explanation) may be more of a factor. Unfortunately, none of the other CESM1 scenarios used in CMIP5 have the exact same forcings as CESM-LENS. For another reviewer, we looked into what criterion was causing the most spread among the CESM-LENS members and found that the EOF maps were far and away the most important for this. Generally, the members were very similar in mean value, trend, and variability, but EOF maps varied significantly. This leads us to believe that the atmospheric forcing is incredibly important in modeling AIS SMB. In all likelihood, both explanations offered by the reviewer play some part in the reality of why there is a notable difference between the LENS project and the other CESM1 members from CMIP5.

**(Suppl.) Figure 6. Put "coarser" and "finer" resolution in figure caption and maybe on figure itself (otherwise easily confusing).**

We have removed figure 6 from the supplementary material at the behest of another reviewer. However, to make sure that this point isn't lost, we have added "coarser" and "finer" to the Figure 5 caption and figure label in the supplementary.

**(Suppl.) Table 1. I think resolution is confusing here. Suggest you make 4 columns: model, resolution (given as lat x lon in traditional manner), total resolution (for correlation calculations eg. Lat/lon), and score. Also many scores are absent?**

We think that adding in a column with the numeral calculation of multiplying the latitude by the longitude will only add confusion. For the figure in which we describe the total resolution, we iterate the y-axis such that it shows perfect square resolutions (i.e. 1x1, 2x2, 3x3, etc). If we were to add the total resolution column, we would change the y-axis to represent simply the product of lat and lon resolutions. While we could do this, we feel that saying something like 1.5 degrees x 1.5 degrees is more intuitive than saying 2.25 degrees sq (especially as most models are close to if not exactly square resolution (i.e. degrees latitude = degrees longitude)). To the reviewer's second point, yes, there appears to have been a formatting error in the manuscript. We have made the requisite formatting changes.

**(Suppl.) Table 4. Perhaps elevated to a table in the main text?**

Presumably the reviewer means Table 5? If so, we have moved the table from supplementary to the main text.

**(Suppl.) What is "GMSL rise buffering"?**

We agree with the reviewer that this phrase is confusion and, as such, have removed the GMSL rise buffering from this table (which has been moved to the main text).

**(Suppl.) Gt yr-2 should be G yr-1 in column one for change in SMB in time**

As SMB is already in units of Gt yr-1, changes to SMB over time would be in units of Gt yr-2.

**(Suppl.) Figure 4. The yellow dot for CESM FASTCHEM is extremely difficult to see (at first I thought it was missing) – for some reason even more difficult in this figure than in Figure 3 in the main text (although hard there too). Suggest changing the color scale a bit or outlining this dot in another color or something to make it visible.**

We agree that the color scale makes it difficult to see that dot. As such, we have darkened the color of the FASTCHEM dot as well as increased the size of the dot.

**Reviewer 2**

**Previous comment 1.1: Comparing GCMs with reconstructions.**
*I am satisfied that analysis of HighResMIP not actually suitable due to the uncoupled nature and shorter duration. I am also satisfied that resolution seems to explain a small amount of inter-model variance in SMB values (just 16%). I would be happy to leave it at that, but the authors seem to contradict themselves on this point. Specifically, their analysis of the range of resolutions across the CMIP5 and CMIP6 models shows a statistically significant relationship between resolution and SMB (Fig. 5 of supplementary material). They state in their response to reviewers that 'This result is further exemplified when looking at total scores from the same model run at different resolutions.' This would seem to strengthen the importance of resolution as a factor that should be considered. The authors then seem to contradict themselves by stating that 'we conclude that there is no significant correlation between model resolution and total Score', which is based on splitting the population into high and low resolution subsets in Figure 6 of supplementary information. I don't follow the rationale for doing this, in particular 6b seems to show a collection of models with almost identical grid spacing. My suggestion is to leave out Figure 6, and use Figure 5 to show that the resolution effect is, although statistically significant, relatively small.*

We will remove Figure 6 from the supplementary material and removed from the main text: "Because so many models… resolution and total score." We appreciate the reviewer's comment about the confusion that our analysis generates and that our response was self-contradictory. There is, indeed, a small yet statistically significant negative correlation between resolution and total score and we have removed text from the supplementary and main text that would muddle that result. We believe that by removing Figure 6 and the associated while making the requisite changes to the remaining text, we have reduced the confusion that it generated.

**Previous comment 1.2: A lack of mechanistic explanation for why each of the 5 criteria are relevant for improving reliability of projections**
*Not really. No clear explanation of each criteria. Leave-one-out can help to see how much difference an evaluation makes to projections. In a response to a later point below the authors state that "In our analysis, we make the significant assumption that the past ability to capture SMB correlates to higher skill in projecting AIS SMB into the future." If the authors follow the above suggestion this need not to be an assumption. It would not be too much work to test whether past ability to capture SMB correlates to higher skill in projecting AIS SMB by looking at projections and biases across the different models to quantify the extent to which they may be correlated.*

We appreciate the reviewer's ideas here about the leave-one-out cross validation (LOOCV) technique. We have looked into this technique and we argue that this technique is not applicable for the purpose of determining "whether past ability to capture SMB correlates to higher skill in projecting AIS SMB." Our understanding is that this LOOCV technique would allow us to determine the ability of using certain criterion scores to predict the total score which does not pertain to the future scenarios. We also argue that the equal weighting of the criteria and the

removal from consideration of the outliers makes this cross-validation moot. The correlations between the criterion scores and the total score and there is no statistically significant difference between the criteria (see table below). We do sincerely apologize if we are misunderstanding the reviewer's point here. We are eager to do additional analysis if we feel it would significantly bolster the rigor of this work which we feel has been evidenced by our analysis of the CESM-LENS experiment and the addition of the CMIP6 SSP scenarios.

| Correlation Variables | score_1 : score_final | score_2 : score_final | score_3 : score_final | score_4 : score_final | score_5 : score_final |
|---|---|---|---|---|---|
| R | 0.51 | 0.64 | 0.53 | 0.42 | 0.48 |
| 95% CI | +/- 0.15 | +/- 0.11 | +/- 0.14 | +/- 0.16 | +/- 0.15 |

**Previous comment 1.3 The methodological framework for model weighting.**
*i. A point that still isn't clear is the consequences of situations with small observational uncertainty. The author's response states that "In regards to the point wherein the reconstruction uncertainty approaches zero, if this is the case, then all the models would score highly on this criterion, that is correct." However, my interpretation is that most models would score badly and one or two models might score very well if they happened to be within one standard deviation of the reconstruction uncertainty. Writing out an equation would really help to clarify.*
*ii. Related to this I'm still not clear on what is meant by "a model that fits entirely within the reconstruction uncertainty" (in 2nd para of Section 3.1). This (I think) needs to be written down mathematically to really clarify. In addition, I couldn't see a clear explanation of the reconstruction uncertainty of AIS-integrated SMB? Is it 5th-95th percentile range for example?*
i. The equation to represent a model score for the SMB mean value criterion would follow as

$$score = \frac{\sum_{1}^{t}\{\ \overline{x(t)} \mid SMB(t) \leq x \cdot uncertainty(t)\}}{t}$$

where t is the year and x is increased incrementally from 1 until the expression $SMB(t) \leq x \cdot uncertainty(t)$ is satisfied. For the remaining two temporal criteria, the expression is simply

$$score = \{\ x \mid \frac{dSMB}{dt} \leq x \cdot uncertainty\}.$$

While this is not actually the case with any of the criteria that we use in our analysis, there could be a possible outcome wherein the uncertainty is large enough that some small subset of models would score well but small enough that the rest would all score very badly. As a part of the previous round of revisions, we added in the stipulation that scores that fall outside of the interquartile range for any criterion would be assigned a maximum value of 10. If the case described above were to actually happen, the interquartile range would be zero and all models would receive a score of 10. If this were the case, that particular criterion would then have no impact on the rest of the overall scoring.

ii. The mathematical expressions for "fitting within the reconstruction uncertainty" are shown above. We do not feel that adding this equation to the main text will add much clarity beyond the figure and words that already appear. As for the reconstruction uncertainty explanation, we had in our previous manuscript a more detailed description of the uncertainty calculation process. However, at the behest of another reviewer, we removed large parts of that section as it was simply rehashing work that had previously been published by Medley & Thomas (2019).

**Previous comment 1.4 The role of internal climate variability in trend and spatial EOF analysis.**
**Probably the most major concern is that the role of internal variability still isn't comprehensively addressed. This is of fundamental importance to the success of the method and must be addressed. Two key aspects that illustrate this are:**
*i. The results of the CESM LENS analysis show a very large range in scoring values for different ensemble members of the same model. This indicates a very strong element of chance (associated with internal variability) in the selection of the top subset (now 8) models, which can be inferred from inspection of Figure 7 (I can see this referenced in the main text). This leads on to the second aspect -*
*ii. It appears to me that the role of internal climate variability is likely significantly under-estimated in the reconstruction uncertainty of the criteria involving trends and EOFs. The real world can be thought of as one ensemble member and therefore the trends and patterns of variability could include a strong element of chance. One approach that is used in detection and attribution studies is to use models to estimate internal climate variability, since there often aren't other alternatives. Essentially, even a perfect reconstruction of what happened in the real world would not necessarily represent the forced climate response and additional uncertainty associated with internal climate variability is still a major consideration. The authors should separate these out into observational uncertainty and internal variability uncertainty.*
i. Yes, we do note in the text that the spread due to internal variability is substantial. We also note that, because some models have numerous ensemble members while others only have one, model means are not really comparable. We have added more to the text to stress this point:
At the end of the Methods section, we have added: "To look at the impact of resolution and internal variability on the final scoring we correlated the horizontal resolution to final score and applied the same scoring analysis to the CESM Large Ensemble (CESM-LENS) experiment."
At the end of the Discussion section, we have also added: "Another significant caveat of this work is the use of single ensemble members. For this work, we use the first ensemble member for each model. This choice was made as the various model members of CMIP5 and CMIP6 vary widely in the number of ensemble members available -- ranging from 1 to 50 -- so using only a single ensemble members helps account for this large disparity between the models. However, in looking at the CESM-LENS experiment -- which has 35 ensemble members -- it is clear that there can be a large spread caused solely by internal variability. The spread in final score among the CESM-LENS ensemble members is 4.65 which is largely generated by the difference in EOF maps meaning that the precise realization of atmospheric conditions in the models is incredibly significant in how the model, in turn, represents AIS SMB."

ii. We are confused here about what is meant by the reconstruction under-estimating internal variability. The reconstruction is based on observations, for which there can be no internal variability (as the reviewer notes: observations are simply one realization of a chaotic climate system) and the reanalysis product MERRA-2, for which there is also only one realization. To that end, there can be no internal variability in the reconstruction. There is uncertainty that is caused by the uncertainty in the ice cores and likely some error that will be generated by the correlation method used to create the product itself (though the latter is difficult to test as there are large swaths of the ice sheet with no data to compare). This uncertainty, though, is incomparable to internal variability and so we are a bit confused about the point that the reviewer seems to be making.

**Reviewer 3**

*Warning: because of my very limited number for working hours these past weeks/months, I only focused on the answers to my comments and to the associated text. I apologize I haven't taken time to re-read the full article, despite my great interest for this work.*

We can greatly appreciate the difficulty with work schedules and thank the reviewer for taking the time to still read and respond as they have.

**To the first point, EOFs map the spatial pattern of a variable associated with the highest temporal variance of another variable**

*Are you sure that is true? From https://www.sciencedirect.com/topics/earth-and-planetarysciences/empirical-orthogonal-function-analysis: "EOFs of a space-time physical process can represent mutually orthogonal space patterns where the data variance is concentrated, with the first pattern being responsible for the largest part of the variance, the second for the largest part of the remaining variance, and so on."*

*It seems that EOF maps spatial patterns associated with total (space-time) variance.*

We appreciate the reviewer's input to make sure the EOF description is as accurate as possible. We have changed the language in the main text (L187-L191 in the marked document) to: "To do so, we performed an empirical orthogonal function (EOF) analysis on annual AIS SMB data from 1850-2000. EOF analysis maps the spatial pattern of a variable where the first mode represents the largest explained variance, the second mode - which is orthogonal to the first - represents the next largest explained variance, the third mode - which is orthogonal to both modes one and two - represents the third largest explained variance, and so on until all the variance is explained."

**in this case, we map the spatial pattern of sea level pressure associated to the highest variability in SMB integrated over the AIS. To the second point, the typo will be corrected.**

*Is there a typo? Is it spatial pattern of SMB instead of spatial pattern of sea level pressure?*

We believe this comment has been addressed in the previous revision. The above comment also further corrects the language of this section.

**Because CMIP6 uses a different future forcing scenario mechanism (Shared Socioeconomic Pathways), CMIP5 and CMIP6 future projections are not directly comparable.**

*SSPs were designed to be comparable to CMIP5 RCPs, with indicated RCP in the SSPs: see Fig.2 of https://www.geosci-model-dev.net/9/3461/2016/gmd-9-3461-2016.pdf. Comparison between CMIP5 and CMIP6 is fully included in the next IPCC report.*

*So you should include CMIP6 in your improved projections.*

We understand what the reviewer is saying here and we see the value in adding comparable analysis for CMIP6. To that end, we have performed the same future analysis on the SSP1-2.6, 2-4.5, and 5-8.5 scenarios so that they can be directly compared to the RCP2.6, 4.5, and 8.5

analyses done for CMIP5. We have changed figures 8-10 accordingly and made numerous additions to the text to reflect this new analysis (too many to specifically list here).

**the top 90th percentile overall scoring models were determined to be GISS E2 H CC, GISS E2 R CC, GISS E2 R, MPI ESM LR, MPI ESM MR, and MPI ESM P**

*It's not unexpected that from the 6 "best" models, there are 2 clusters of same modeling center simulations: 3 GISS-E2 and 3 MPI-ESM. I think you must add a criteria in your model selection to select models from different modeling centers, because these models will share the same biaises (see e.g. https://www.pnas.org/content/115/38/9462). I think it is important to sample a diversity in model simulations. Can you remove 2 GISS and 2 MPI and add 2 to 4 other models instead?*

*If you don't, you should add a table with all models ranked by the final score and including the score for each criteria and modeled SMB projection for each scenario; ~similarly to what you did in Supplementary but including all criteria scores and projections.*

*NB: it seems that Tables in Supplementary materials are not correctly displayed*

We appreciate what the reviewer is saying here about modeling center diversity. We feel that the consistency in modeling center is a demonstrable effect of the preeminent role of model physics in determining AIS SMB. As such, we are leaving much of the main text as is. However, we do not want to disregard the comment as there is relevance to knowing information akin to the best scoring modeling centers. To that point, we have added Figures 6 and 7 to the supplementary in a new section titled "Modeling Centers" with the associated text: "Of the eight best scoring models, seven originate from two modeling centers: the Max Planck Institute fur Meteorologie (MPI) and Goddard Institute for Space Studies (GISS) from NASA. This strongly implies that model physics plays a significant role in the representation of AIS SMB. Another interpretation could be, though, that these models simply share the same biases and, thus, all are coincidentally favorably compared to the reconstruction. Here, we also look at a more diverse spread of modeling centers in two ways: 1) the top eight models that originate from unique modeling centers and 2) the top four modeling centers (top 90th percentile) averaged across their members. Figures 6 and 7 show the best scoring eight models from unique modeling centers and best four scoring modeling centers on average, respectively. The former category consists of GISS R, MPI ESM LR, CESM2 FV2, FGOALS G2, MIROC ESM, INM CM4, IPSL CM5A MR, and ACCESS ESM1-5 (Fig. 6). MPI ESM, GISS, FGOALS, and INM CM from CMIP5 constitute the latter category of best modeling centers on average (Fig. 7).."

**These eight models have been added in retroactively to figures 2-3**

*I don't see added models in Fig. 2? Maybe it's Fig. 3 and Fig. 4?*

This typo has been corrected to "figures 3-5."

**The reconstructed AIS SMB averaged from 1801-2000 shows Along with higher SMB values around the coastal areas, particularly in the Antarctic Peninsula and West Antarctic regions (Fig. 2A). The highest absolute SMB trends are around the , the coastal regions of East Antarctica and the Antarctic Peninsula also show the highest absolute SMB trends (Fig. 2B). This reconstruction also highlights large portions of...**

*You have to introduce the reconstruction in the modified sentence*
We are not entirely sure what the reviewer is saying here. We believe this text was removed during the previous round of revision and that the comment no longer applies.

---

## Author Response (AR3)

[revised manuscript text omitted]

**I think they are almost there in terms of addressing the issue of internal variability and the Figure below is great to see. I would just ask that the Figure is reproduced for a different statistic(s). Instead of the standard deviation of inter-annual variability, the statistics used for scoring the models should be shown for different numbers of ensemble members. For example, for CESM-LENS the y-axis would be the raw SMB trend (1920-2005 for CESM-LENS and 1851-2000 if repeated for the CMIP5 CESM runs) for the first ensemble member, then the second, and so on. This should be done for all five criteria. The inter-ensemble-member variability of the criteria would give a model estimate of how large the internal variability term might be in the equation: observational uncertainty = reconstruction uncertainty + internal variability**
**One way to think about this is that if the scoring is well calibrated, then different ensemble members of the same model should not exhibit a wide range of different scores.**
**In summary, if the authors re-produce Figure 1 for the different scoring criteria and assess the implications of this for the role of internal variability in the observational uncertainty I think there is a clear route to publication.**

We thank the editor and reviewer again for all their efforts with this paper. To address these comments, we have done the analysis described below.
We calculated the standard deviation in raw value among CESM LENS ensemble members for the three ice sheet integrated scoring criteria to represent 'intra-ensemble variability' as the reviewer requested (Fig 1). We then used this standard deviation to calculate the observational uncertainty using the equation

observational uncertainty = sqrt(reconstruction uncertainty^2 + intra-ensemble variability^2).

We then rescored all the CMIP5/6 models using this new observational uncertainty instead of the reconstruction uncertainty. We believe this should take into account the interensemble variability to which the reviewer refers. These new results clearly show that the impact of considering intra-ensemble variability does not substantially change the overall observational uncertainty, the latter which remains largely dominated by the reconstruction uncertainty.

[Figure]

**Figure 1:** Panels show the raw values for each ensemble member in CESM LENS. The left panel shows the temporal average of AIS-integrated SMB, the center panel shows AIS-integrated SMB trend, and the right panel shows AIS-integrated SMB variability. The dashed line indicates the average across all ensemble members and the shaded region denotes 1-sigma (one standard deviation). At the bottom of each panel is the reconstructed uncertainty value for each criterion and the newly calculated observational uncertainty.

We would like to repeat our earlier statement, i.e. that a similar analysis is not possible to perform for the EOF analyses, because of two reasons: (1) we see no reliable way of introducing this 'intra-ensemble' variability in a time-varying spatial map of SMB that is used for EOF analysis; (2) as mentioned in earlier responses, even though we would add 'noise' to that field, the EOF patterns would not change, only their magnitudes.

We hope that this analysis addresses the remaining issues by the reviewer. We are happy to add this figure (as well as the figure suggested in the previous response), to the manuscript, and add some discussion. Please let us know if you have any questions or need us to elaborate further on anything we've discussed in this email. Again, we thank you and the reviewer for your hard work and diligence.

---

## Author Response (AR4)

[revised manuscript text omitted]
. Unlike Figure 7 in the Results section of the main text, this figure highlights the eight best scoring models from unique modeling centers (i.e. no one modeling center can have more than one of the top eight models) which are denoted with red outlines if they are among the CMIP5 suite of models or with blue outlines if they are among the CMIP6 suite of models. The eight best modeling centers, then, are GISS, MPI, FGOALS, MIROC, INM, and IPSL from CMIP5 and CESM2 and ACCESS from CMIP6.

**7  Data Tables**

60  This section includes tables with model resolutions and scores for all CMIP5 and CMIP6 models as well as a table with projected SMB and related variables for the various RCPs.

[Figure]

**Figure 8.** The scores for all CMIP5 and CMIP6 models. The dots show the average score for all model groupings. Models are grouped by similar model physics and have in parentheses the number of models in the grouping after the name. Here, the top scoring 4 modeling centers (90th percentile of 41 modeling centers) on average are denoted with red outlines as they are all within the CMIP5 suite of models. The four best modeling centers on average are MPI, GISS, FGOALS, MIROC, and INM from CMIP5.

|   | Model | Resolution (°lat×°lon) | Total Score |
|---|---|---|---|
| 1 | ACCESS1-0 | 1.2414×1.875 | 3.80 |
| 2 | ACCESS1-3 | 1.2414×1.875 | 3.38 |
| 3 | BCC-CSM1.1 | 2.8125×2.8125 | 4.85 |
| 4 | BCC-CSM1.1-m | 2.8125×2.8125 | 4.99 |
| 5 | BNU ESM | 2.8125×2.8125 | 10 |
| 6 | CanESM2 | 2.8125×2.8125 | 5.24 |
| 7 | CCSM4 | 0.9375×1.25 | 7.40 |
| 8 | CESM1 BGC | 0.9375×1.25 | 5.81 |
| 9 | CESM1 CAM5 FV2 | 0.9375×1.25 | 3.99 |
| 10 | CESM1 CAM5 | 0.9375×1.25 | 5.30 |
| 11 | CESM1 FASTCHEM | 0.9375×1.25 | 9.74 |
| 12 | CESM1 WACCM | 0.9375×1.25 | 6.63 |

**Table 1.** Model names, resolutions and final score for the first half of the CMIP5 suite of models.

| | Model | Resolution (°lat×°lon) | Total Score |
|---|---|---|---|
| 13 | CMCC CESM | 0.75×0.75 | 3.94 |
| 14 | CMCC CM | 0.75×0.75 | 3.50 |
| 15 | CNRM CM5 | 1.4063×1.4063 | 4.73 |
| 16 | CSIRO | 1.875×1.875 | 3.98 |
| 17 | FGOALS | 3×2.8125 | 2.07 |
| 18 | FIO ESM | 2.8125×2.8125 | 8.89 |
| 19 | GFDL CM3 | 2×2.5 | 3.83 |
| 20 | GFDL ESM2G | 2×2.5 | 4.48 |
| 21 | GFDL ESM2M | 2×2.5 | 7.11 |
| 22 | GISS E2 H CC | 2×2.5 | 1.72 |
| 23 | GISS E2 H | 2×2.5 | 3.55 |
| 24 | GISS E2 R CC | 2×2.5 | 1.60 |
| 25 | GISS E2 R | 2×2.5 | 1.00  |
| 26 | HadGEM2 CC | 1.2414×1.875 | 5.14 |
| 27 | HadGEM2 ES | 1.2414×1.875 | 5.56 |
| 28 | INMCM4 | 1.5×2 | 2.61 |
| 29 | IPSL CM5A LR | 1.875×3.75 | 3.13 |
| 30 | IPSL CM5A MR | 1.2587×2.5 | 2.88 |
| 31 | IPSL CM5B LR | 1.875×3.75 | 5.50 |
| 32 | MIROC ESM CHEM | 1.4063×1.4063 | 2.75 |
| 33 | MIROC ESM | 2.8125×2.8125 | 2.86 |
| 34 | MIROC5 | 2.8125×2.8125 | 2.19 |
| 35 | MPI ESM LR | 1.875×1.875 | 1.59 |
| 36 | MPI ESM MR | 1.067×1.067 | 1.76 |
| 37 | MPI ESM P | 1.875×1.875 | 1.76 |
| 38 | MRI CGCM3 | 1.125×1.125 | 3.92 |
| 39 | MRI ESM1 | 1.125×1.125 | 4.46 |
| 40 | NorESM1 M | 1.875×2.5 | 3.89 |
| 41 | NorESM1 ME | 1.875×2.5 | 4.30 |

65  **Table 2.** Model names, resolutions and final score for the second half of the CMIP5 suite of models.

|  | Model | Resolution (°lat×°lon) | Total Score |
|---|---|---|---|
| 1 | ACCESS CM2 | 1.25×1.875 | 3.66 |
| 2 | ACCESS ESM1-5 | 1.25×1.875 | 3.25 |
| 3 | BCC CSM2 MR | 1.125×1.125 | 7.17 |
| 4 | BCC ESM1 | 2.8125×2.8125 | 6.20 |
| 5 | CAMS CSM1 | 1.125×1.125 | 8.78 |
| 6 | CanESM5 | 2.8125×2.8125 | 8.25 |
| 7 | CanESM5-CanOE | 2.8125×2.8125 | 4.42 |
| 8 | CESM2 | 0.9375×1.25 | 8.04 |
| 9 | CESM2 FV2 | 1.9×2.5 | 2.08 |
| 10 | CESM2 WACCM | 0.9375×1.25 | 6.93 |
| 11 | CESM2 WACCM FV2 | 1.9×2.5 | 3.76 |
| 12 | CNRM CM6-1 | 1.4063×1.4063 | 6.63 |
| 13 | CNRM CM6-1-HR | 0.5×0.5 | 8.10 |
| 14 | CNRM ESM2 | 1.4063×1.4063 | 6.09 |
| 15 | E3SM1 | 1.0×1.0 | 8.79 |
| 16 | E3SM1-1 | 1.0×1.0 | 5.17 |
| 17 | E3SM1-1 ECA | 1.0×1.0 | 5.09 |
| 18 | FGOALS F3 L | 2.0×2.25 | 5.88 |
| 19 | FGOALS G3 | 2.0×2.25 | 5.72 |
| 20 | GFDL ESM4 | 1.0×1.25 | 7.71 |
| 21 | GISS E2 G | 2.0×2.5 | 5.81 |
| 22 | GISS E2 G CC | 2.0×2.5 | 3.55 |
| 23 | GISS E2 H | 2.0×2.5 | 6.78 |
| 24 | HadGEM3 GC3 | 1.25×1.875 | 7.97 |
| 25 | IPSL CM6A | 1.2587×2.5 | 6.30 |
| 26 | INM CM4-8 | 1.5×2.0 | 3.66 |
| 27 | INM CM5-0 | 1.5×2.0 | 4.04 |
| 28 | KACE1-0-G | 1.25×1.875 | 4.78 |
| 29 | MCM UA1 | 2.25×3.75 | 8.42 |
| 30 | MIROC6 | 2.8125×2.8125 | 7.14 |
| 31 | MIROC E2SL | 2.8125×2.8125 | 5.38 |

**Table 3.** Model names, resolutions and final score for the first half of CMIP6 suite of models.

| | Model | Resolution (°lat×°lon) | Total Score |
|---|---|---|---|
| 32 | MPI ESM1-2-HAM | 1.875×1.875 | 2.16 |
| 33 | MPI ESM1-2-HR | 0.935×0.935 | 8.04 |
| 34 | MPI ESM1-2-LR | 1.875×1.875 | 1.77 |
| 35 | MRI ESM2 | 1.125×1.125 | 7.16 |
| 36 | NESM3 | 1.875×1.875 | 3.45 |
| 37 | NorCPM1 | 1.875×2.5 | 8.00 |
| 38 | NorESM2-MM | 0.9375×1.25 | 3.80 |
| 39 | SAM0 UNICON | 0.9375×1.25 | 7.58 |
| 40 | UKESM1 | 0.9375×1.875 | 8.45 |

**Table 4.** Model names, resolutions and final score for the second half of CMIP6 suite of models.

70 **8 AIS Map**

Figure 9 shows the AIS with the names of locations specifically mentioned in the main text.

[Figure]

**Figure 9.** Map of the AIS.

The authors have gone a long way to addressing the final concern on the role of internal variability in the observational uncertainty. There are however a few more clarifications on the edits that would be required before recommending acceptance. The review classification of 'major revision' is mainly a reflection that the below comments are important in terms of correctly communicating the rationale for the reformulated observational uncertainty calculation (and could potentially involve some revision of figures), but ultimately if they are addressed I fully expect to be able to recommend acceptance. It would be a great help if the authors could respond clearly to each one of the below comments in turn.

The comments below relate to two broad points: (i) The explanation of internal variability in the observational uncertainty needs to be developed more. (ii) I was surprised that the figures and scores (specifically Figures 1, 3, 4, 7 and supplementary Table 1) all look identical to the previous version. If they have been calculated to include ensemble (natural) variability I would have expected some very slight changes at least. It would be useful if the authors could show some examples of the each term in the scoring before and after the inclusion of internal variability.

Specific points:

1. Line 77: The statement
- "the intra-ensemble variability takes into account the internal variability of climate models"
needs to be revised to something like
- "the intra-ensemble variability uses climate models to estimate uncertainty in observations due to internal variability of the climate system".
This is required because the point of the new term in the equation is to capture the fact the observational uncertainty is a combination of reconstruction uncertainty and internal climate variability (estimated here from the LENS ensemble).

At line 77, we have changed "The inclusion of the intra-ensemble variability takes into account the internal variability of climate models." to "The inclusion of the internal variability uses the spread generated from climate models to estimate uncertainty in observations due to internal variability of the climate system."

2. The explanation relating to the above text should to reference other literature, for example Tokarska et al. (2020) that I mentioned in my previous round of revisions. The authors are welcome to use some of the text from this previous review to help expand the explanation.

After the added sentence "The inclusion of the internal variability uses the spread generated from climate models to estimate uncertainty in observations due to internal variability of the climate system," we have also added "Tokarska et al. (2020) note that noise due to internal variability can be derived using a finite number of model ensemble members."

3. The notation of equation 1 needs to be changed to reflect the above point. For example as I suggested in my previous round of revisions:
"uncertainty = reconstruction uncertainty + internal variability"

Maybe it would be clearer as follows:
"observational uncertainty = reconstruction uncertainty + internal variability"

We have changed the notation throughout to reflect the following:

total uncertainty = reconstruction uncertainty + internal variability uncertainty

4. Apologies if I'm missing something, but it's not clear what is shown in the numbers in blue in supplementary Figure 5. The average and 1-sigma range in 5B looks like it's about 1.95 +/- 0.35 Gt yr^-2. How does this relate to the numbers displayed ("obs. unc. = 1.33 Gt yr^-2" and "recon. unc. = 1.38 Gt yr^-2")?

Yes, those numbers are roughly correct. To account for accurate error propagation, we added uncertainties in quadrature:

sqrt((1.33 Gt yr^-2)^2 + (0.35 Gt yr^-2)) ≈ 1.38 Gt yr^-2.

Perhaps the original chosen notation was confusing. As noted above, we have changed the notation so the new Figure 5B would read "recon. unc. = 1.33 Gt yr^-2, total unc. = 1.38 Gt yr^-2."

5. More generally, from Figure 5B in particular the internal variability aspect looks quite large. What is needed is each of the three terms of equation 1 shown explicitly in each panel, which would greatly help in clarifying and comparing the different terms.

We have added the third term of the equation to the figure.

6. Figure 7 and 8 of the supplementary material are missing captions.

We apologize for this oversight. We have added to Figure 7 in the supplementary the caption: "The scores for all CMIP5 and CeMIP6 models. The large dots show the average score for all model groupings. Models are grouped by similar model physics and have in parentheses the number of models in the grouping after the name. Each model grouping has all model scores plotted as small blue/red dots for CMIP5/6 with the model average plotted in the larger dots. Models that have no like models are followed by a one in parentheses and only have a larger dot. Unlike Figure 7 in the Results section of the main text, this figure highlights the eight best scoring models from unique modeling centers (i.e. no one modeling center can have more than one of the top eight models) which are denoted with red outlines if they are among the CMIP5 suite of models or with blue outlines if they are among the CMIP6 suite of models. The eight best modeling centers, then, are GISS, MPI, FGOALS, MIROC, INM, and IPSL from CMIP5 and CESM2 and ACCESS from CMIP6." To Figure 8 in supplementary, we have added "The scores for all CMIP5 and CMIP6 models. The  dots show the average score for all model groupings. Models are grouped by similar model physics and have in parentheses the number of models in the grouping after the name. Here, the top scoring 4 modeling centers (90th percentile of 41 modeling centers) on average are denoted with red outlines as they are all within the CMIP5 suite of models. The four best modeling centers on average are MPI, GISS, FGOALS, MIROC, and INM from CMIP5."